# Improving Rationality in the Reasoning Process of Language Models through Self-playing Game

**Pinzheng Wang** [1 2]  **Juntao Li** [1 2]  **Zecheng Tang** [1 2]  **Haijia Gui** [1 2]  **Min Zhang** [1 2]

## Abstract

Large language models (LLMs) have demonstrated considerable reasoning abilities in various tasks such as mathematics and coding. However, recent studies indicate that even the best models lack true comprehension of their reasoning processes. In this paper, we explore how self-play can enhance the rationality of models in the reasoning process without supervision from humans or superior models. We design a *Critic-Discernment Game* (**CDG**) in which a prover first provides a solution to a given problem and is subsequently challenged by critiques of its solution. These critiques either aim to assist or mislead the prover. The objective of the prover is to maintain the correct answer when faced with misleading comments, while correcting errors in response to constructive feedback. Our experiments on tasks involving mathematical reasoning, stepwise error detection, self-correction, and long-chain reasoning demonstrate that CDG training can significantly improve the ability of well-aligned LLMs to comprehend their reasoning process. We have released our code here.

## 1. Introduction

Large language models (LLMs) have achieved significant success in reasoning tasks such as coding and mathematics (Roziere et al., 2023; Dubey et al., 2024; Yang et al., 2024a). By training on reasoning paths from humans or superior models, LLMs can generate impressive step-by-step reasoning processes similar to human thought (Kim et al., 2023; Wei et al., 2022). However, recent studies indicate that even well fine-tuned and aligned LLMs still lack true

comprehension of their reasoning processes and instead rely mostly on probabilistic pattern matching (Mirzadeh et al., 2024; Valmeekam et al., 2022; Razeghi et al., 2022). This can be reflected in the instability of the reasoning process of LLMs, which tends to produce hallucinations and errors while struggling to detect and correct these issues on its own (Pan et al., 2023; Kamoi et al., 2024). This limitation is particularly detrimental for long chains of thought, which are essential for solving complex and challenging tasks. As the length of the reasoning process increases, intermediate errors and incorrect attempts are more likely to accumulate if the model lacks the ability to detect or correct them (Zhao et al., 2024; Wu et al., 2024a).

To mitigate this issue, existing approaches employ process based reward models (PRMs) or preference data pairs to provide stepwise error supervision for the reasoning process (Lightman et al., 2023; Lai et al., 2024; Uesato et al., 2022a). However, these methods struggle to explicitly define fine-grained steps in general reasoning, merely indicate which step is better without providing specific explanations, and frequently rely on human-annotated data, making them difficult to scale up (Guo et al., 2025; Zhang et al., 2025). Therefore, we explore enhancing the rationality of the reasoning process of LLMs through self-play in language, without supervision from humans or superior models.

We design a *Critic-Discernment Game* (**CDG**), where the agent engages in discussion about its reasoning process with different goals. Specifically, our game involves three roles: the prover, the helpful critic, and the misleading critic. As shown in Figure 1, each role has the following objective:

- **Prover**: Given a question, the prover is required to provide an answer with a clear chain of thought. Afterward, the prover receives feedback from a critic whose intent can be either helpful or misleading. The prover should maintain its initial correct solution and revise the incorrect solution.

- **Helpful Critic**: The helpful critic receives a problem along with the prover's incorrect answer. Its task is to identify the mistake in the prover's reasoning without directly correcting the solution, persuading and assisting the prover in revising the original incorrect answer.

- **Misleading Critic**: The misleading critic receives a prob-

[1]School of Computer Science and Technology, Soochow University [2]Key Laboratory of Data Intelligence and Advanced Computing, Soochow University. Correspondence to: Juntao Li <ljt@suda.edu.cn>.

*Proceedings of the 42$^{nd}$ International Conference on Machine Learning*, Vancouver, Canada. PMLR 267, 2025. Copyright 2025 by the author(s).

lem and the correct solution of the prover. Its task is to fabricate and point out a false error within the prover's solution, intending to mislead the prover to change the original correct answer into an incorrect one.

The critics can freely choose the granularity of the steps they critique, either to attack or to collaborate with the prover in the form of natural language. All three roles are optimized through reinforcement learning (RL) algorithms based on their respective tasks and performance in each round. To achieve success, **the prover must thoroughly understand its reasoning process, ensuring it can maintain correct steps without being misled by the increasingly adaptive misleading critic, while accurately revising genuinely erroneous steps.**

To the best of our knowledge, we are the first to improve reasoning capabilities through a self-play language game on fully fine-tuned and aligned models, such as LLaMA-3.1-8B-instruct (Dubey et al., 2024) and Qwen2.5-1.5B-Instruct (Yang et al., 2024a). To evaluate whether the model truly gains a better understanding of its reasoning process, we conduct comprehensive evaluations across four tasks: mathematical reasoning, stepwise error detection, self-correction, and long-chain reasoning. In the long-chain reasoning task, the model is required to generate reasoning processes similar to OpenAI-o1 (OpenO1 Team, 2024), which involves extensive trial, self-reflection, and error correction to solve hard math problems (Wu et al., 2024a; Qin et al., 2024; Huang et al., 2024). Our method consistently improves model performance across all tasks, demonstrating that the Critic-Discernment Game effectively enhances rationality in the reasoning process. We believe our method highlights the potential of self-play language games with RL as a novel training paradigm beyond instruction tuning and preference optimization (Rafailov et al., 2024; Schulman et al., 2017; Meng et al., 2024).

## 2. Related works

**RL for LLM reasoning**  Reinforcement Learning (RL) has emerged as a crucial training method to enhance reasoning capabilities in large language models (LLMs), with many methods providing feedback based on the quality of the reasoning outcome, measured by the correctness of the final answer (Anthony et al., 2017; Gulcehre et al., 2023; Singh et al., 2023) or through an outcome reward model (ORM) (Cobbe et al., 2021c; Yang et al., 2024b; Uesato et al., 2022b). However, recent works have highlighted that LLMs lack true comprehension of their reasoning processes, making them prone to generating hallucinations and errors (Mirzadeh et al., 2024; Lanham et al., 2023). To address this, many works have introduced step-level supervision for models, usually through process reward models (PRMs) (Zhang et al., 2025; Uesato et al., 2022a) or

step-wise preference data (Lai et al., 2024). These methods, however, rely on data annotated by humans or superior models, which limits their scalability. Some methods attempt to automate the construction of step-level supervision using techniques like Monte Carlo Tree Search (Zhang et al.; Lu et al., 2024; Wang et al., 2024; Luo et al., 2024). Yet, these approaches struggle to explicitly define fine-grained reasoning steps and tend to have limitations of data quality issues or reward hacking (Guo et al., 2025). In this work, we propose a self-play RL algorithm which differs from traditional preference optimization. In our approach, rewards are derived from game rules, allowing for step-level supervision at arbitrary granularity without relying on human or superior model supervision.

**Self-Play in LLM**  Self-play is a technique where multiple agents learn by interacting with themselves (Zhang et al., 2024; Cheng et al., 2024a; Xi et al., 2024). Game agents like AlphaGo Zero (Silver et al., 2017) have shown that advanced planning and reasoning abilities can be developed solely through self-play without human supervision. Inspired by these successes, many efforts have been made to enhance various LLM capabilities via self-play games. Zheng et al. (2024) and Xu et al. (2024) employ a two-agent attack-defense game to align LLMs in safety-critical scenarios, significantly improving robustness against jailbreak attacks. Additionally, many works (Chen et al., 2024; Wu et al., 2024b; Kirchner et al., 2024) utilize self-play strategies, such as generate-discriminate frameworks and competitive reward mechanisms, to better align LLMs with human preferences. Furthermore, studies have shown that self-play can effectively enhance agents' strategies in language games, e.g., Negotiation (Fu et al., 2023) and the Werewolf Game (Xu et al., 2023a;b). Our work is most similar to SPAG (Cheng et al., 2024b), which uses the Adversarial Taboo Game to improve pre-trained models' reasoning abilities. However, our approach differs in that we explore how self-play can enhance the rationality of the reasoning process in well-instructed and aligned models, such as Llama-3.1-8B-Instruct, rather than merely activating latent abilities in pre-trained models.

## 3. Self-play of Critic-Discernment Games

### 3.1. Critic-Discernment Game Modeling

In our game, we consider a dataset consisting of math questions and their corresponding ground truth answers $y = y(x)$. Given a solution $z$ proposed for a question $x$, we employ regular expressions and the SymPy grader (Meurer et al., 2017) to evaluate the correctness of the solution, denoted by $\mathbb{1}_{\text{correct}}(z, y)$. We represent the prover, helpful critic, and misleading critic as $\pi$, $\mu$, and $\rho$, respectively.

At the start of the game, the prover is required to provide

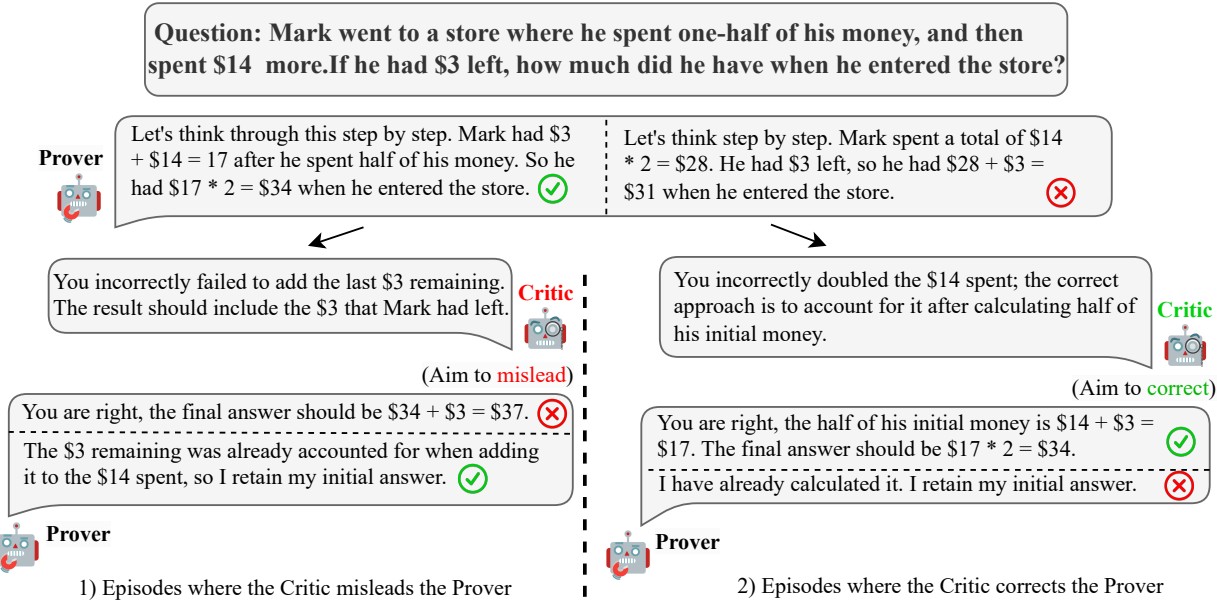

Figure 1. Examples of Critic-Discernment Game with the same math question. In the left dialogue, the critic attempts to mislead the prover into changing a correct answer, while in the right dialogue, the critic guides the prover to correct an incorrect answer. The prover must rationally decide whether to revise their response based on the critic's feedback and their own answer.

---

**Algorithm 1** Data collection of Critic-Discernment Game

**Inputs:** prover $\pi$, helpful critic $\mu$, misleading critic $\rho$, question $x$ and corresponding answer $y$, prompt templates $f_{\text{help}}$, $f_{\text{mislead}}$, $f_{\text{revise}}$

Sample prover's initial solution $z \sim \pi(z|x)$

**if** $\mathbb{1}_{\text{correct}}(z_\pi, y) = 0$ **then**

  Sample critique $c \sim \mu(c_\mu|f_{\text{help}}(x, z))$

**else**

  Sample critique $c \sim \rho(c_\rho|f_{\text{mislead}}(x, z))$

**end if**

Sample revised solution $z'_\pi \sim \pi(z_\pi|f_{\text{revise}}(x, z, c))$

Collect an episode $\tau = (x, z, c, z')$

---

an initial solution $z$ for a given problem sampled from the dataset distribution $P_x$. If the solution $z$ is incorrect, the game assigns a helpful critic to identify and highlight the error in the solution; conversely, if the solution $z$ is correct, the game assigns a misleading critic to fabricate a non-existent error in the solution, attempting to mislead the prover into revising the correct answer to an incorrect one. Upon receiving critiques whose intent is unknown, the prover must decide whether to modify the solution by reassessing the highlighted errors in its reasoning process. The data collection procedure for a single Critic-Discernment Game is described in Algorithm 1. The prompt templates $f_{\text{help}}$, $f_{\text{mislead}}$, $f_{\text{revise}}$ used in the game are provided in Appendix A. The prover can win the game in two ways: (1) by providing

a correct solution on the first attempt and successfully resisting the misleading critic's deception, or (2) by initially providing an incorrect solution but correcting it through feedback from the helpful critic to arrive at the correct final answer. The first way of winning requires a superior ability to solve the problem, and therefore we assign a larger reward to this outcome. The total reward for the prover is:

$$R_\pi = \mathbb{E}_{z\sim\pi}[\underbrace{\mathbb{1}_{\text{correct}}(z, y)\mathbb{E}_{c\sim\rho, z'\sim\pi}(\mathbb{1}_{\text{correct}}(z', y) + \eta)}_{\text{(1) Correct initially and resist misleading critic}}$$
$$+ \underbrace{(1 - \mathbb{1}_{\text{correct}}(z, y))\mathbb{E}_{c\sim\mu, z'\sim\pi}\mathbb{1}_{\text{correct}}(z', y)}_{\text{(2) Incorrect initially but correct with the help of the helpful critic}}],$$
(1)

where $\eta$ is the hyper-parameter to ensure a larger reward of the first term, $z \sim \pi(z|x)$, $z' \sim \pi(z'|f_{\text{revise}}(x, z, c))$, $c \sim \mu(c|f_{\text{help}}(x, z))$, $c \sim \rho(c|f_{\text{mislead}}(x, z))$ are simplified as $z \sim \pi$, $z' \sim \pi$, $c \sim \mu$, $c \sim \rho$.

For the helpful critic, the condition for winning is to convince the prover of its critique and guide it in successfully correcting its initial incorrect solution. The reward for the helpful critic is assigned as follows:

$$R_\mu = \mathbb{E}_{z\sim\pi, c\sim\mu, z'\sim\pi}[(1 - \mathbb{1}_{\text{correct}}(z, y))\mathbb{1}_{\text{correct}}(z', y)].$$
(2)

For the misleading critic, the condition for winning is to successfully deceive the prover into altering a correct solution

based on its critique. The reward is assigned as follows:

$$R_\rho = \mathbb{E}_{z\sim\pi, c\sim\rho, z'\sim\pi}[\mathbb{1}_{\text{correct}}(z, y)(1 - \mathbb{1}_{\text{correct}}(z', y))]. \quad (3)$$

Each role in the game is trained to maximize its own reward. The prover and the helpful critic work collaboratively, while the prover competes against the misleading critic. During the training process, we jointly optimize the strategies of these models through reinforcement learning.

### 3.2. Reinforcement Learning from Self-play

To enhance the capabilities of each role in the game, we employ reinforcement learning (RL) for training. The prover model is optimized to maximize the expected reward value:

$$\mathcal{L}_{\text{RL-CDG}}(\pi) = -\mathbb{E}\big[R_\pi(\pi(z_\pi|x_\pi))\big], \quad (4)$$

where $z_\pi$ denotes the output of the prover given input $x_\pi$.

Several mainstream reinforcement learning algorithms can be employed to optimize the above objective, including Proximal Policy Optimization (PPO), Direct Preference Optimization (DPO), and Reinforced Self-Training (ReST). Our experiments show that among these methods, ReST exhibits stable performance and considerable effectiveness in our critic-discernment game. Therefore, we adopt ReST as the training method for CDG agents. With a threshold $\tau \in \mathbb{R}$, ReST updates the LLM by the reinforcement on the selected samples $\mathcal{D}_{\tau_\pi} = \big\{(x_\pi, z_\pi) : r(x_\pi, z_\pi) > \tau_\pi\big\}$. The corresponding loss function can be written as:

$$\begin{aligned}\mathcal{L}_{\text{ReST-CDG}}(\pi) &= \mathbb{E}\big[\mathbb{1}_{r(x_\pi, z_\pi)>\tau_\pi}\mathcal{L}_{\text{LM}}(x_\pi, z_\pi)\big] \\ &= \mathbb{E}_{\mathcal{D}_{\tau_\pi}}[\mathcal{L}_{\text{LM}}(x_\pi, z_\pi)], \quad (5)\end{aligned}$$

where $\mathcal{L}_{\text{LM}}(x_\pi, z_\pi)$ denotes the language modeling loss on the self-generated data.

The loss function for critic $\mu$ and $\rho$ can be defined as:

$$\mathcal{L}_{\text{ReST-CDG}}(\rho) = \mathbb{E}_{\mathcal{D}_{\tau_\rho}}[\mathcal{L}_{\text{LM}}(x_\rho, z_\rho)] \quad (6)$$

$$\mathcal{L}_{\text{ReST-CDG}}(\mu) = \mathbb{E}_{\mathcal{D}_{\tau_\mu}}[\mathcal{L}_{\text{LM}}(x_\mu, z_\mu)]. \quad (7)$$

In our experiments, the values of $\tau_\pi$, $\tau_\rho$, and $\tau_\mu$ are set to $0.5$, $0.75$, and $0.5$, respectively. According to equation 1, the samples included in $\mathcal{D}_{\tau_\pi}$ for reinforcement are: (1) cases where the prover provides a correct solution on the first attempt; (2) cases where the prover successfully resists misleading critiques; and (3) cases where the prover successfully corrects the initial incorrect solution with the assistance of the helpful critic. For the critics, each critique is presented to the prover in the form of $f_{\text{revise}}(x, z, c)$, and the prover is required to generate multiple responses through random sampling. A misleading critique is included in $\mathcal{D}_{\tau_\rho}$ only if it successfully misleads the prover with a success

---

**Algorithm 2** Self-play of Critic-Discernment Game

**Inputs:** inital prover $\pi_1$, initial helpful critic $\mu_1$, initial misleading critic $\rho_1$, iteration number for self-improvement $T$

**for** iteration $t = 1$ to $T$ **do**
  Collect self-play episodes $\mathcal{T}_t = \{\tau \sim \pi_t \times \mu_t \times \rho_t\}$
  Select $\mathcal{D}^t_{\tau_\pi}, \mathcal{D}^t_{\tau_\mu}, \mathcal{D}^t_{\tau_\rho}$ from $\mathcal{T}_t$
  **if** $t > 1$ **then**
    $\mathcal{D}^t_{\tau_\pi} \leftarrow \mathcal{D}^t_{\tau_\pi} \cup \mathcal{D}^{t-1}_{\tau_\pi}$
    $\mathcal{D}^t_{\tau_\mu} \leftarrow \mathcal{D}^t_{\tau_\mu} \cup \mathcal{D}^{t-1}_{\tau_\mu}$
    $\mathcal{D}^t_{\tau_\rho} \leftarrow \mathcal{D}^t_{\tau_\rho} \cup \mathcal{D}^{t-1}_{\tau_\rho}$
  **end if**
  Minimize the following objective on $\pi_1$ to obtain $\pi_{t+1}$:
  $\mathcal{L}_{\text{ReST-CDG}}(\pi) = \mathbb{E}_{\mathcal{D}^t_{\tau_\pi}}[\mathcal{L}_{\text{LM}}(x_\pi, z_\pi)]$
  Minimize the following objective on $\mu_1$ to obtain $\mu_{t+1}$:
  $\mathcal{L}_{\text{ReST-CDG}}(\mu) = \mathbb{E}_{\mathcal{D}^t_{\tau_\mu}}[\mathcal{L}_{\text{LM}}(x_\mu, z_\mu)]$
  Minimize the following objective on $\rho_1$ to obtain $\rho_{t+1}$:
  $\mathcal{L}_{\text{ReST-CDG}}(\rho) = \mathbb{E}_{\mathcal{D}^t_{\tau_\rho}}[\mathcal{L}_{\text{LM}}(x_\rho, z_\rho)]$
**end for**

---

rate exceeding $\tau_\rho$. Similarly, a helpful critique is included in $\mathcal{D}\tau_\mu$ only if it helps the prover correct an initially incorrect solution with a success rate exceeding $\tau_\mu$.

Considering that the feature of multi-turn auto-regressive generation by LLMs is inefficient for on-policy RL training, we adopt an offline learning scheme for multi-round training. For the $t$-th training round, we first collect self-play episodes and update the datasets for training each role as $D^t_{\tau_{role}} = D^{t-1}_{\tau_{role}} \cup D^t_{\tau_{role}}$. Subsequently, we update the initial policy model by minimizing Equations 5, 6 and 7, and use it as the initialization for the next round of self-play. More details of our self-play strategy are provided in Appendix B. The overall training algorithm is described in Algorithm 2.

## 4. Experiments

To evaluate the effectiveness of CDG, we conduct experiments on four mathematics-related reasoning tasks using the fully fine-tuned LLaMA3.1-8B-Instruct model. These tasks are designed to comprehensively assess the model's rationality in its reasoning process. Notably, LLaMA3.1-8B-Instruct has already undergone self-improvement using techniques such as rejection sampling and Monte Carlo Tree Search with reward guidance for mathematical tasks (Dubey et al., 2024). Therefore, performance improvements observed from training with CDG demonstrate its ability to further enhance the model's reasoning rationality on the foundation of traditional self-improvement methods.

### 4.1. Improving Mathematical Reasoning with CDG

In this section, we evaluate the mathematical reasoning capabilities of various models under the CDG framework.

**Improving Rationality in the Reasoning Process of Language Models through Self-playing Game**

*Table 1.* Math reasoning performance of CDG, which consistently enhances the reasoning ability of each model (p-value $< 0.05$). CDG-$n$ represents the prover model trained after the $n$-th iteration, while CDG-Imitation refers to the model trained using imitation learning.

| Models | Methods | GSM8K | | | MATH500 | | |
|---|---|---|---|---|---|---|---|
| | | $P@1$ | $M@8$ | $M@32$ | $P@1$ | $M@8$ | $M@32$ |
| **Instruction-tuned Models** | | | | | | | |
| Llama-3.1-8B-Instruct | None | 85.3 | 91.2 | 93.0 | 49.4 | 58.3 | 63.4 |
| | CDG-1 | 85.5 | 91.1 | 92.7 | 49.0 | 56.8 | 62.2 |
| | CDG-2 | **86.8** | **92.0** | **93.1** | **51.7** | **60.7** | **66.0** |
| Qwen2.5-1.5B-Instruct | None | 75.1 | 82.6 | 86.5 | 55.4 | 58.6 | 62.9 |
| | CDG-1 | **75.4** | **83.8** | 86.4 | **57.6** | **62.9** | **66.2** |
| | CDG-2 | 75.2 | **83.8** | **86.8** | 56.1 | 61.5 | 65.6 |
| **Pre-trained Model** | | | | | | | |
| Llama-3.1-8B-Base | + CDG-Imitation | 78.9 | 87.1 | 90.4 | 29.4 | 36.5 | 42.6 |
| | CDG-1 | 78.4 | 86.4 | 89.3 | 29.8 | 36.3 | 40.4 |
| | CDG-2 | **79.2** | **88.7** | **90.8** | **33.8** | **39.2** | **43.6** |

**Datasets and Evaluation Metrics**   We focus on the field of mathematical reasoning using two widely used datasets: GSM8K and MATH500 (Cobbe et al., 2021a;b). The training sets contain 7,473 and 12,000 samples, respectively, while the test sets consist of 1,319 and 500 samples. The problem-answer pairs for self-play training are sourced from their respective training sets. We evaluate all models using greedy sampling (Pass@1) and majority voting with 8 and 32 samples (M@8, M@32).

**Backbone Models**   To investigate the impact of self-play on different sizes of models, we select two instruction-tuned models: Llama3.1-8B-Instruct (Dubey et al., 2024) and Qwen2.5-1.5B-Instruct (Yang et al., 2024a) as backbone models. Additionally, to explore whether self-play can improve the performance of pre-trained models that have not undergone instruction tuning, we also conduct experiments on Llama-3.1-8B-Base.

**CDG Training**   For instruction-tuned models, we follow Algorithm 2 and perform two rounds of self-play training. For dataset $\mathcal{D}_{\tau_\pi}$ selected during training, we ensure that the number of samples where the prover correctly answer on the first attempt, successfully resist misleading critiques, and correctly revise errors are balanced at 10,000 each. For pre-trained models, we introduce an additional imitation learning step before self-play to establish basic game-playing capabilities and ensure outputs following the self-play format. The imitation learning dataset, consisting of 30,000 examples, is generated by Llama-3.1-8B-Instruct. All evaluations are conducted on the prover model. Further details in CDG training can be found in Appendix B.

**Results**   As shown in Table 1, CDG training consistently enhances mathematical reasoning capabilities across all models and datasets. In particular, for both Llama-3.1-8B-Instruct and Llama-3.1-8B-Base, the second round of training significantly improves the prover's reasoning performance. Case study on generated episodes reveals that this improvement is primarily due to the stronger attacks from the RL-trained misleading critic, which forces the prover to gain a deeper understanding of its reasoning process during the second training round, as shown in Appendix D. Furthermore, our method achieves larger improvements on the MATH500 dataset, highlighting CDG's ability to enhance the models' performance on more challenging problems.

### 4.2. Enhancing Stepwise Error Detection in the Reasoning Process with CDG

**Task**   We evaluate the model's understanding of its own reasoning by testing its ability to identify potential errors in its reasoning steps. Concretely, the model receives a question and solution and is tasked with examining a specified step to determine its correctness. Notably, we evaluate the model on problems it has already demonstrated proficiency in solving, ensuring that the assessment focuses solely on its ability to recognize errors in the reasoning process.

**Test Data Construction**   To ensure that the model's stepwise error detection ability is evaluated on problems it is capable of solving, we include only problems for which the majority of sampled solutions are correct in the test set. We then employ GPT-4o to identify the first erroneous step in the incorrect solutions for these problems. Through this process, we construct samples consisting of solutions paired with their corresponding erroneous steps. Similarly, we ran-

domly sample one step from the model's correct solutions to create samples where each of the solutions is paired with a correct step. The final dataset contains 200 positive samples and 200 negative samples.

**Results** Table 2 presents the accuracy and $F_1$ score for the stepwise error detection task. The results show that CDG training significantly improves error detection in reasoning, particularly on the more challenging MATH500 dataset, enhancing the model's self-reflection ability.

*Table 2.* Performance of CDG on stepwise error detection.

| Models | GSM8K | | MATH500 | |
|---|---|---|---|---|
| | $F_1$ | Acc | $F_1$ | Acc |
| Llama-3.1-8B-Instruct | 74.0 | 64.4 | 64.4 | 55.4 |
| CDG | 76.9 | 69.3 | 71.4 | 67.5 |

## 4.3. Improving Self-Correction with CDG

**Task** To further investigate whether CDG Training can enhance a model's self-awareness of its reasoning steps and improve the rationality of its reasoning process, we evaluate Llama3.1-8B-Instruct on the Self-Correction task. The task is structured as follows: the model first generates an initial response to a given question. Subsequently, the user prompts the model with the command, "Please check your answer step by step again", instructing the model to review and correct its initial response. Previous studies have indicated that models exhibit very limited self-correction abilities without external feedback (Kamoi et al., 2024; Pan et al., 2023). They tend to misinterpret their reasoning process and mistakenly modify initially correct answers, resulting in a degraded final response.

**Self-Correction Training** To equip the model with foundational self-correction capabilities, we fine-tune Llama3.1-8B-Instruct both before and after CDG training on identical datasets. Specifically, we use the vanilla Llama3.1-8B-Instruct to generate eight responses for each question in the training set and then label them as correct or incorrect based on the ground truth. Following this, the model conducts a second round of self-checking and correction to improve its initial responses. We construct a self-correction training set consisting of two types of examples: 5,000 cases where the model's initial correct answers are maintained after reflection in the second round, and 5,000 cases where initial erroneous answers are corrected to accurate ones. This self-correction dataset is then used to fine-tune the model. We provide more details in Appendix B.

**Results** As illustrated in Figure 2, the model trained by CDG significantly reduces the rate at which initial correct

answers are modified erroneously during self-correction, while maintaining a nearly unchanged rate of correcting incorrect answers successfully, even when the initial response accuracy generated by CDG is higher. In GSM8K, the model trained with CDG reduces the probability of erroneously modifying correct responses to less than half that of the original model. In general, CDG training maintains a positive and improved self-correction rate compared to the vanilla model. These results suggest that the model achieves a more rational evaluation of its reasoning process after CDG training.

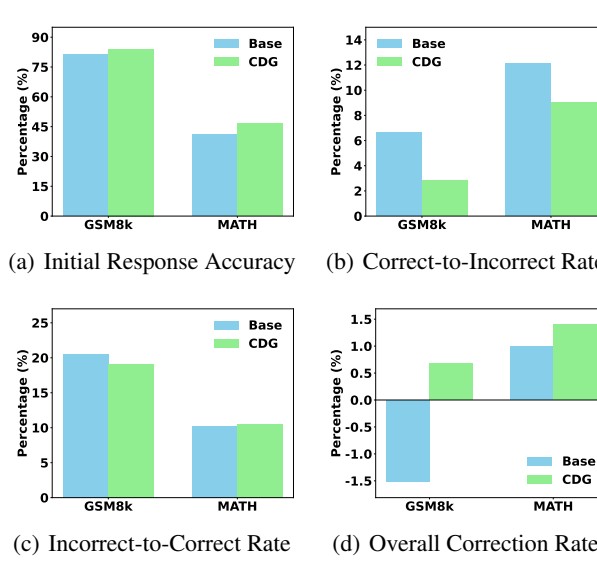

(a) Initial Response Accuracy  (b) Correct-to-Incorrect Rate

(c) Incorrect-to-Correct Rate  (d) Overall Correction Rate

*Figure 2.* Self-Correction results before and after CDG training.

## 4.4. Improving Long-Chain Reasoning with CDG

**Task** OpenAI-o1 demonstrates that long-chain reasoning, which involves extensive exploration, self-reflection, and self-correction, can significantly enhance the ability to solve complex reasoning problems by increasing test-time compute (Wu et al., 2024a; Zhao et al., 2024), which is similar to how humans tackle difficult problems. However, such reasoning requires the LLM to have a strong understanding of its own reasoning process, enabling it to identify when to correct errors and adjust its exploration direction. We test LLM performance on challenging problems from MATH500, focusing on level 5 difficulty, under the long-chain reasoning paradigm.

**Long-Chain Reasoning Training** To endow the model with long-chain reasoning capabilities, we distill the reasoning process of QwQ-32B-Preview (Team, 2024) into Llama3.1-8B-Instruct, both before and after CDG training. QwQ-32B-Preview is an open-source LLM renowned for solving problems through extensive reasoning, questioning, and reflection, with an average reasoning chain length

*Table 3.* Performance comparison of other RL methods and variants of CDG.

| Method | Math Reasoning | | Error Detection | | Self Correction | | Long-chain Reasoning | Average |
|---|---|---|---|---|---|---|---|---|
| | GSM8K | MATH | GSM8K | MATH | GSM8K | MATH | | |
| Llama-3.1-8B-Instruct | 85.3 | 49.4 | 74.0 | 64.4 | -1.5 | 1.0 | 25.3 | 42.5 |
| **Other RL Methods** | | | | | | | | |
| Expert Iteration | **87.2** | 50.8 | 75.4 | 67.4 | -0.6 | 0.8 | 22.8 | 43.4 |
| Step-DPO | 84.6 | 51.6 | 69.8 | 58.2 | -2.2 | -2.1 | 27.6 | 41.0 |
| **CDG and its variants** | | | | | | | | |
| CDG | 86.8 | **51.7** | **76.9** | **71.4** | **0.7** | **1.4** | **29.7** | **45.5** |
| CDG w/o Correct critic | 86.2 | 50.9 | 76.2 | 69.7 | -0.5 | -3.0 | 28.3 | 43.9 |
| CDG w/o Misleading critic | 84.9 | 51.1 | 76.2 | 68.9 | -1.2 | -0.5 | 29.4 | 44.1 |

significantly exceeding that of Llama3.1-8B-Instruct.

Directly distilling the reasoning chains of QwQ-32B-Preview into Llama3.1-8B-Instruct, however, results in unstable performance due to the capability gap between the two models. To mitigate this, we first perform supervised fine-tuning (SFT) on Llama3.1-8B-Instruct using reasoning chains generated by QwQ-32B-Preview. Subsequently, we apply rejection sampling based on ground truth answers to filter the reasoning chains generated by Llama3.1-8B-Instruct post-SFT. Compared to direct distillation, this filtered dataset is more suitable for enhancing the capabilities of Llama3.1-8B-Instruct. We then use this dataset to train both pre- and post-CDG training versions of the model for Long-Chain Reasoning tests.

**Results**  As shown in Figure 3, the two-stage distillation significantly enhances the original model's performance on challenging mathematical problems, achieving nearly a 10-point improvement. Additionally, models trained with CDG demonstrate superior long-chain reasoning capabilities compared to the original model when distilled on the same dataset, showing a consistent 3-5 point improvement. This indicates that CDG training enables models to develop better long-chain reasoning abilities, reflecting a more rational understanding of their own reasoning steps.

## 5. Analysis of the Critic-Discernment Game

### 5.1. Comparing With Other RL Methods

To better assess the effectiveness of CDG training in enhancing model capabilities, we compare it with two mainstream RL approaches: Expert Iteration (Anthony et al., 2017) and StepDPO (Lai et al., 2024). Expert Iteration is one of the most effective RL-based methods for improving reasoning in LLMs (Havrilla et al., 2024). It fine-tunes models on high-return responses using standard cross-entropy loss. Following Havrilla et al. (2024), we train the model exclu-

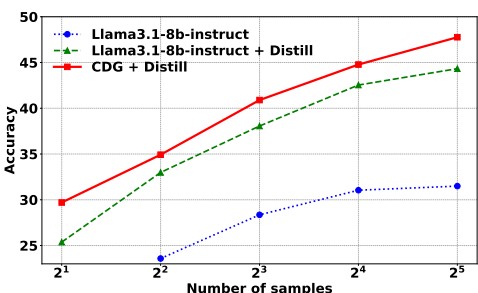

*Figure 3.* Performance on Level-5 Difficulty of MATH. The x-axis represents the number of samples, while the y-axis shows the accuracy under majority voting. Models trained with CDG demonstrate significantly better long-chain reasoning capabilities compared to models without CDG training, with a p-value $< 0.05$.

sively on responses that yield correct final answers for two iterations under the same training budget as CDG. StepDPO improves reasoning by using GPT-4o (Hurst et al., 2024) to generate a step-level pairwise preference dataset, which is then used for training via DPO. We follow the experimental setup from the original paper and conduct training on LLaMA 3.1-8B-Instruct using the same dataset.

As shown in Table 3, Expert Iteration, which relies solely on outcome rewards, significantly improves performance on the Math Reasoning task but performs poorly on Long-chain Reasoning. In contrast, StepDPO, which applies step-wise supervision, enhances Long-chain Reasoning but underperforms in Error Detection and Self-Correction. Compared to these methods, our approach consistently enhances performance over the vanilla model across all four tasks, achieving superior results in Error Detection, Self-Correction, and Long-chain Reasoning, while performing on par with Expert Iteration in Math Reasoning. These results demonstrate the effectiveness of CDG training in improving the rationality of the reasoning process.

## 5.2. Component Analysis

We conduct experiments on two variants of CDG training to investigate the key factors contributing to the prover's improvement in our Critic-Discernment Game. In these variants, either only the helpful critic or only the misleading critic interacts with the prover, while maintaining the same objective as in the vanilla CDG training. As shown in Table 3, the two variants of CDG training perform worse across all tasks compared to the full CDG training. Notably, these variants even lead to a decline in self-correction ability compared to the vanilla Llama3.1-8B-Instruct. This suggests that the prover, when trained in a single-critic setting, may become either overly reliant on the critic or completely disregard it. Such tendencies may hinder the prover from genuinely understanding its reasoning process. Therefore, the effectiveness of CDG training stems not only from the ability to correct initial errors or resist misleading critiques, but also from the prover's ability to discern the true quality of its reasoning process.

## 5.3. Self-play Training with Various RL Methods

We also investigate the impact of applying different RL algorithms in CDG training. Specifically, we experiment with two mainstream RL methods: Direct Preference Optimization (DPO) (Rafailov et al., 2024) and Proximal Policy Optimization (PPO) (Schulman et al., 2017). All three algorithms are trained using the same set of collected game episodes. To construct pairwise preference data for DPO, we generate responses to the same prompt by randomly sampling outputs with rewards of 1 (chosen) and 0 (rejected) for the prover. For PPO, we follow the experimental setup of SPAG (Cheng et al., 2024b), employing the same method for advantage estimation. Additional training details for DPO and PPO are provided in Appendix B.

Table 4 presents the performance of CDG with various RL methods on math reasoning. ReST consistently achieves the best performance under both greedy decoding and majority voting settings. In contrast, DPO even underperforms the vanilla LLaMA-3.1-8B-Instruct model. PPO improves reasoning performance in the greedy decoding setting but degrades performance under majority voting. This degradation may be attributed to the sensitivity of these algorithms to hyperparameter choices (Havrilla et al., 2024). Compared to these methods, the ReST method achieves more stable performance improvements while requiring less GPU memory, as it eliminates the need for a reference model.

## 5.4. Game-play Performance

Besides evaluating the mathematical reasoning abilities of LLMs, we also assess the model's performance in the game by analyzing its win rate on the test set. We conduct exper-

*Table 4.* Performance of CDG with various reinforcement learning methods on math reasoning.

| Models | GSM8K | | MATH500 | |
|---|---|---|---|---|
| | P@1 | M@32 | P@1 | M@32 |
| Llama-3.1-8B-Instruct | 85.3 | 93.0 | 49.4 | 63.4 |
| CDG-ReST | 86.8 | 93.1 | 51.7 | 66.0 |
| CDG-DPO | 83.3 | 92.5 | 46.0 | 54.8 |
| CDG-PPO | 86.6 | 92.9 | 51.6 | 62.6 |

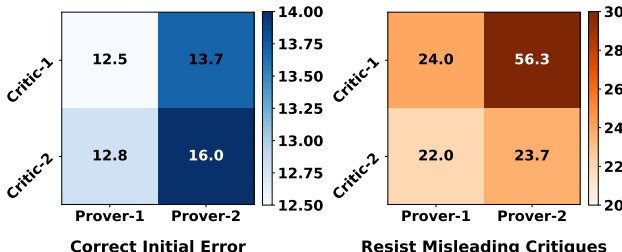

*Figure 4.* Game results on the Critic-Discernment Game. Left: Average win rates of prover models with different training stages when collaborating with critic models. Right: Average win rates of prover models in resisting critiques from the misleading critic.

iments where the prover from different training iterations interacts with helpful critics and misleading critics from different training iterations.

As shown in Figure 4, as training progresses, both the prover and the helpful critic improve their ability to identify and correct errors, achieving their best performance in the final training iteration. This aligns with their cooperative learning objectives. Additionally, we observe that the prover in loop 2 is most effective at resisting the misleading critic in loop 1, while the prover in loop 1 performs the worst against the misleading critic in loop 2. This result is consistent with their adversarial optimization objectives, where the prover and the misleading critic compete against each other. We can also find that if the misleading critic is not sufficiently trained, the prover in the second training loop can easily resist it, with the win rate increasing from 24.0 to 56.3. This is because the initial misleading critic lacks sufficient game-playing capability, which aligns with our observation that a single round of CDG provides limited or even no improvement in math reasoning as shown in Table 1.

## 6. Conclusion

We introduce a novel training strategy: Self-play learning in the Critic-Discernment Game, to effectively enhance the rationality of LLMs in the reasoning process. In our framework, a prover is challenged by critiques of its solution,

which may either be helpful or misleading. The task of the prover is rationally deciding whether to revise their response based on the critiques. Through multiple epochs of self-play and reinforcement learning with ReST, we observe that LLMs exhibit consistent improvements in reasoning performance across various tasks, demonstrating a deeper understanding of their own reasoning processes. We believe that our method highlights the potential of self-play language games with RL as a promising new training paradigm for enhancing the reasoning capabilities of LLMs.

## Impact Statement

This paper presents work aimed at advancing the field of reasoning with large language models. As with all research involving large language models, there are inherent risks, including the spread of misinformation and the propagation of societal biases.

## Acknowledgements

We want to thank all the anonymous reviewers for their valuable comments. This work was supported by the National Science Foundation of China (NSFC No. 62206194), the Natural Science Foundation of Jiangsu Province, China (Grant No. BK20220488), the Young Elite Scientists Sponsorship Program by CAST (2023QNRC001), and the Priority Academic Program Development of Jiangsu Higher Education Institutions.

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

## A. Data Collection Details

**Self-play of CDG**   Figure 5, Figure 6, and Figure 7 illustrate the prompt templates used in the Critic-Discernment Game (CDG). To ensure consistency in the critique styles generated by the misleading critic and the helpful critic to prevent potential reward hacking, we employ identical prompts for both types of critics. During the process of collecting episodes for self-play training, both the prover and the critic generate responses with a temperature of 0.95, a maximum length of 4096, a top-p of 0.95, and a top-k of 5.

---

**Question Template**

**User:** Solve the following math problem efficiently and clearly:
Regardless of the approach, always conclude with: Therefore, the final answer is: $\boxed{answer}$. I hope it is correct. Where [answer] is just the final number or expression that solves the problem.
Here is the question you need to answer:
#### Question: {question}

**Assistant:**

---

*Figure 5.* Prompt template for answering mathematical questions.

---

**Critic Template**

**User:** Please critic the answer carefully.
Your task is to evaluate a question-answer pair.
Carefully review the question and critically assess a wrong answer.
[Question]
```
{question}
```
[Wrong Answer]
```
{solution}
```
Please review the wrong answer step by step, quoting each sentence and analyzing it individually. Use the following format for your response:
Step: [Quoted Sentence]
Analysis: [Your Explanation]
Step: [Quoted Sentence]
Analysis: [Your Explanation]
After step-by-step analysis, conclude by quoting the original sentence which **first** causes the wrong answer and providing a concise yet complete explanation of the error:
"**Critic** The first mistake can be found in: 'Quoted wrong statement here.' The issue is: 'Explanation of the mistake here.'"

**Assistant:**

---

*Figure 6.* Prompt template for generating critics.

---

**Step-wise Error Detection**   Figure 8 presents the prompt used to annotate erroneous steps with gpt-4o. If the generated solution is not explicitly divided into steps, we segment it based on "\n\n". Figure 9 illustrates the prompt designed to evaluate the model's stepwise error detection capability.

**Self-Correction**   Figure 10 illustrates the prompt used for self-correction.

---

**Revising Template**

**User:** {question}

**Assistant:** {initial solution}

**User:** Please check with this critic. "{critic}"
Evaluate whether this critic is valid. Keep in mind that this critic might be misleading or irrelevant.
If you find the critic incorrect, conclude with: "\boxed{This critic is not critical.}"
If you determine the critic to be valid, revise starting from the incorrect step, and present your revised answer within \boxed{}.

**Assistant:**

---

*Figure 7.* Prompt template for revising answers based on critiques.

---

**Error Detection Template (gpt-4o)**

**User:** Question:
{question}
Wrong Response:
{solution}
Identify the first step that contains an error. Please provide the step number in the format: Step \boxed{X}

**Assistant:**

---

*Figure 8.* Prompt template for detecting stepwise error by gpt-4o.

---

**Error Detection Template**

**User:** Solve the following math problem efficiently and clearly:
Regardless of the approach, always conclude with: Therefore, the final answer is: $\boxed{answer}$. I hope it is correct. Where [answer] is just the final number or expression that solves the problem.
Here is the question you need to answer:
#### Question: {question}

**Assistant:**

---

*Figure 9.* Prompt template for evaluating stepwise error detection.

---

**Self-Correction Template**

**User:** {question}

**Assistant:** {initial response}

**User:** Please check your answer step by step again. Put your final answer within \boxed{}.
**Assistant:**

---

*Figure 10.* Prompt template for self correction.

## B. Self-play training Details

To generate game episodes for reinforcement learning (RL) training, we require the prover to generate four initial responses upon receiving a math problem. Then, the helpful critic generates eight critiques for each question-solution pair, while the misleading critic generates four critiques for each question-solution pair. Subsequently, for each triplet of (question, solution, critique), the prover produces four responses.

**Training CDG with ReST**   When constructing the dataset $\mathcal{D}_{\tau_\pi}$ for ReST training, we determine correctness using regular expressions and the SymPy grader (Meurer et al., 2017). A prover's response is considered correct either if it provides the correct solution in the first round or successfully correct an initially incorrect solution in the second round. To evaluate whether the prover successfully resists misleading critiques, we check whether the model outputs "\boxed{This critic is not critical.}".

For constructing $\mathcal{D}_{\tau_\mu}$, a helpful critic's critique is included only if the corresponding prover generates at least two correct responses for the given (question, solution, critique) triplet. For constructing $\mathcal{D}_{\tau_\rho}$, a misleading critic's critique is included only if the corresponding prover fails to output "\boxed{This critic is not critical.}" and changes a previously correct answer into an incorrect one in at least three responses for the same (question, solution, critique) triplet.

In the first training loop, we use a learning rate of 5e-6 and a batch size of 32 to facilitate rapid convergence. In the second loop, as the dataset size increases, we adjust the learning rate to 1e-6 and the batch size to 256. The prover and misleading critic are trained for one epoch, while the helpful critic is trained for two epochs.

**Training CDG with DPO**   In Section 5, we also explore training the Critic-Discernment Game (CDG) using Direct Preference Optimization (DPO) (Rafailov et al., 2024) in a self-play setting. To construct pairwise preference data for training the prover, we select one successful and one unsuccessful response from the prover's generated outputs for each triplet of (question, solution, critique).

We use a learning rate of 1e-6, a batch size of 64, and set $\beta = 0.5$ to control the deviation from the base reference policy. We observe that DPO training for CDG is highly sensitive to hyperparameter selection and often results in repetitive text generation.

**Training CDG with PPO**   In Section 5, we also explore training CDG through Self-Play using PPO (Schulman et al., 2017). For each episode, we assign rewards according to Equations 5, 6, and 7. We then follow the PPO training framework used in SPAG (Cheng et al., 2024b) to conduct self-play training.

We set the learning rate to 5e-6 and the batch size to 128. To control the deviation from the base reference policy, we set $\beta = 0.2$. Additionally, to ensure stability when answering questions on the first attempt, we incorporate a supervised fine-tuning loss with a weighting coefficient of 0.5.

## C. Experiments

In this section, we provide additional details regarding our experimental setup and evaluation.

### C.1. Mathematical Reasoning

For mathematical reasoning tasks, we use GSM8K (Cobbe et al., 2021a) and MATH500 (Lightman et al., 2023) as our test datasets, which contain 1,319 and 500 examples, respectively. Under the Pass@1 setting, we generate solutions using greedy decoding. In the majority voting setting, we set the maximum sequence length to 8,192, the temperature to 0.95, and use top-$k = 10$ for sampling-based generation.

### C.2. Stepwise Error Detection

For the Stepwise Error Detection task, we first generate eight solutions for each test sample using the Llama-3.1-8B-Instruct model with a temperature of 0.95. We include only problems for which the majority of sampled solutions are correct in the test set. When annotating stepwise error with GPT-4o, we use greedy decoding and filter out examples with formatting errors. We use greedy decoding during evaluation.

### C.3. Self-Correction

Compared to Stepwise Error Detection, Self-Correction is a more challenging task, as it requires the model not only to identify errors but also to correctly revise them. To construct the training dataset for self-correction, we first collect solutions generated by Llama-3.1-8B-Instruct with a temperature of 0.95 on the training set. Following this, the model performs a second round of self-checking and correction to refine its initial responses. We select cases where the final answer is correct for training. Additionally, we remove data instances where the self-checking process contains repetitive loops.

Referring to Algorithm 2, our final CDG model is obtained by fine-tuning the original Llama-3.1-8B-Instruct on $\mathcal{D}_{\tau_\pi}^2$. Subsequently, we perform another fine-tuning step on the $\mathcal{D}_{\text{self-correct}}$ dataset. During our experiments, we observed that sequential fine-tuning on two different distributions can lead to catastrophic forgetting (Luo et al., 2023). To mitigate this issue, we directly fine-tune the original Llama-3.1-8B-Instruct on the combined dataset $\mathcal{D}_{\tau_\pi}^2 \cup \mathcal{D}_{\text{self-correct}}$. For the baselines presented in Table 3, we apply the same fine-tuning strategy to ensure a fair comparison. We set the learning rate to $1e-6$ and the batch size to 256.

During testing, we first generate solutions using the model with a temperature of 0.95 on the test set. The model is then required to perform a second round of self-checking and correction on the initial solutions using greedy decoding.

### C.4. Long-Chain Reasoning

In Section 4.4, we introduce our two-stage fine-tuning approach for enhancing long-chain reasoning capabilities. Specifically, we utilize QwQ-32B-Preview (Team, 2024) to generate solutions on the training set with a temperature of 0.95, selecting only those with correct final answers for fine-tuning. Additionally, we filter out examples containing repetitive loops during generation. Similarly, Llama-3.1-8B-Instruct post-SFT generates solutions with a temperature of 0.95, and only those with correct final answers are selected for fine-tuning, while filtering out instances with repetitive loops.

For training long-chain reasoning capabilities in CDG and other methods, we follow the same approach as in the Self-Correction task and merge the training datasets. The learning rate for both fine-tuning stages is set to $1e-6$, and the batch size is 256.

## D. Self-play Examples

We present examples of the first and second rounds where the prover successfully corrects the initial error, successfully resists the misleading critique, and fails to resist the misleading critique in n Figures 11, 12, 13, 14, 15, 16, and 17.

---

**Self-play in loop1 (Prover successfully corrects the initial error)**

**User:** Determine the number of ways to arrange the letters of the word THAT.

**Prover:** To determine the number of ways to arrange the letters of the word THAT, we can use the concept of permutations. Since the word has no repeated letters, the total number of permutations can be calculated using the factorial of the number of letters.
The word THAT has 4 letters (T-H-A-T). Therefore, the total number of permutations is 4! (4 factorial).
$4! = 4 \times 3 \times 2 \times 1 = 24$
Therefore, the final answer is: $\boxed{24}$.

**Assistant:** Please check with this critic.
"The first mistake can be found in: "Since the word has no repeated letters, the total number of permutations can be calculated using the factorial of the number of letters."
The issue is: The presence or absence of repeated letters is not the correct condition to determine whether to use the factorial formula. A correct condition is that the letters should be distinct. In the word "THAT," the letters are not distinct due to the presence of repeated Ts. This mistake leads to an incorrect application of the factorial formula, resulting in an incorrect final answer of 24
Evaluate whether this critic is valid. Keep in mind that this critic might be misleading or irrelevant.
If you find the critic incorrect, conclude with: "\boxed{This critic is not critical.}"
If you determine the critic to be valid, revise starting from the incorrect step, and present your revised answer within \boxed{}.

**Prover:** The critic is pointing out that the condition for using the factorial formula is not the absence of repeated letters, but the distinctness of the letters. In the case of the word "THAT," the presence of two identical "T"s makes the letters not distinct.
Given that the letters in "THAT" are not distinct due to the repeated "T"s, we need to adjust the calculation. When there are repeated elements, we divide the total number of permutations by the factorial of the number of repetitions for each element.
In "THAT," there are 2 "T"s. The correct formula for permutations with repeated elements is:
Total permutations = (Number of letters)! / (Number of repetitions for each letter)!
Total permutations = 4! / (2! for the repeated "T"s)
Now, calculate the total permutations:
$4! = 4 \times 3 \times 2 \times 1 = 24$
$2! = 2 \times 1 = 2$
Total permutations = 24 / 2 = 12
Therefore, the correct answer is: $\boxed{12}$.
This critic is not misleading; it is valid and correct.

---

*Figure 11.* Self-play examples of Llama3.1-8B-Instruct in loop1 (Prover successfully corrects the initial error).

---

**Self-play in loop1 (Prover successfully resists the misleading critique)**

**User:** The distance across a country is 8205 kilometers. Amelia started at one end and drove 907 kilometers on Monday and 582 kilometers on Tuesday. How many kilometers does Amelia still have to drive to make it across the country?

**Assistant:** To find out how many kilometers Amelia still has to drive, we need to subtract the distance she has already driven from the total distance across the country.
First, we need to find the total distance Amelia has driven so far.
- She drove 907 kilometers on Monday.
- She drove 582 kilometers on Tuesday.
So, the total distance she has driven is 907 + 582 = 1489 kilometers.
Now, we subtract the total distance Amelia has driven from the total distance across the country.
- Total distance across the country = 8205 kilometers.
- Distance driven so far = 1489 kilometers.
So, the distance Amelia still has to drive is 8205 - 1489 = 6716 kilometers.
Therefore, the final answer is: $\boxed{6716}$. I hope it is correct.

**User:** Please check with this critic.
"The first mistake can be found in: 'She drove 907 kilometers on Monday.' The issue is: There is actually no mistake in the given wrong answer. However, the original question does not state the order of the distances driven, so the answer assumes that the distances are in the order they are driven. If we assume that Amelia drove 582 kilometers on Monday and 907 kilometers on Tuesday, we get a different result. The correct total distance driven would be 582 + 907 = 1489 kilometers, but this would still be correct since $1489 < 8205$. However, if we consider that the order of the distances could be different, then the answer is actually not wrong in this case."
Evaluate whether this critic is valid. Keep in mind that this critic might be misleading or irrelevant.
If you find the critic incorrect, conclude with: "\boxed{This critic is not critical.}"
If you determine the critic to be valid, revise starting from the incorrect step, and present your revised answer within \boxed{}.

**Assistant:** The critic points out that the original solution assumes the order of the distances driven (Monday and Tuesday). However, the problem statement does not specify the order of the distances.
To address this concern, we need to consider both possibilities:
- Amelia drove 907 kilometers on Monday and 582 kilometers on Tuesday.
- Amelia drove 582 kilometers on Monday and 907 kilometers on Tuesday.
In both cases, the total distance driven would be the same: 1489 kilometers (582 + 907).
However, the critic's main point is that we should be cautious when assuming the order of the distances. But, in this case, it does not change the result. The original solution is correct regardless of the order of the distances.
Since the critic does not provide a valid reason to change the original answer, I would say that:
$\boxed{\text{This critic is not critical.}}$

*Figure 12.* Self-play examples of Llama3.1-8B-Instruct in loop1 (Prover successfully resists the misleading critique).

---

**Self-play in loop1** (Prover fails to resist the misleading critique)

**User:** Find the arithmetic mean of the reciprocals of the first three prime numbers.

**Assistant:** To find the arithmetic mean of the reciprocals of the first three prime numbers, we need to follow these steps:

1. Identify the first three prime numbers, which are 2, 3, and 5.
2. Find the reciprocals of these prime numbers, which are $\frac{1}{2}$, $\frac{1}{3}$, and $\frac{1}{5}$ respectively.
3. Add these reciprocals together: $\frac{1}{2} + \frac{1}{3} + \frac{1}{5}$.
4. Find a common denominator, which in this case is 30. So, the sum becomes:
$\frac{15}{30} + \frac{10}{30} + \frac{6}{30} = \frac{31}{30}$.
5. Divide the sum by 3 to find the arithmetic mean: $\frac{31}{30} \div 3 = \frac{31}{30} \times \frac{1}{3} = \frac{31}{90}$.

Therefore, the final answer is: $\boxed{\dfrac{31}{90}}$.

**User:** Please check with this critic.

"The first mistake can be found in: 'Therefore, the final answer is: $\boxed{\dfrac{31}{90}}$.' The issue is: The final answer is incorrect because it is the reciprocal of the sum of the fractions, which is not the arithmetic mean of the first three prime numbers."
Evaluate whether this critic is valid. Keep in mind that this critic might be misleading or irrelevant.
If you find the critic incorrect, conclude with: "\boxed{This critic is not critical.}"
If you determine the critic to be valid, revise starting from the incorrect step, and present your revised answer within \boxed{}.

**Assistant:** The critic is valid. The arithmetic mean of a set of numbers is found by adding the numbers and dividing by the number of items. In this case, the arithmetic mean of the reciprocals of the first three prime numbers should be found by adding the reciprocals of 2, 3, and 5, and then dividing by 3.

To correct this, we need to revisit step 4 and then proceed to step 5.

1. Identify the first three prime numbers, which are 2, 3, and 5.
2. Find the reciprocals of these prime numbers, which are $\frac{1}{2}$, $\frac{1}{3}$, and $\frac{1}{5}$ respectively.
3. Add these reciprocals together: $\frac{1}{2} + \frac{1}{3} + \frac{1}{5}$.
4. Find a common denominator, which in this case is 30. So, the sum becomes: $\frac{15}{30} + \frac{10}{30} + \frac{6}{30} = \frac{31}{30}$.
5. To find the arithmetic mean, we should divide the sum by the total number of items, which is 3. However, we can directly take the sum, $\frac{31}{30}$, then divide that by 3 to find the mean of the 3 numbers $\frac{1}{2}$, $\frac{1}{3}$, and $\frac{1}{5}$.

The correct arithmetic mean should be found as: $\frac{\frac{31}{30}}{3} = \frac{31}{90}$.

However, we are finding the mean of the reciprocals of the first three prime numbers. Therefore, to get the actual mean of the reciprocals, we divide the sum by 3.

$\frac{\frac{31}{30}}{3} = \frac{1}{30} \cdot \frac{31}{1} \cdot \frac{1}{3} = \frac{31}{90}$ is correct in that the mean of the three fractions is $\frac{1}{2} + \frac{1}{3} + \frac{1}{5}$ which is $\frac{31}{30}$, but to find the mean of these 3 reciprocals, we must divide $\frac{31}{30}$ by 3.

Therefore, the correct arithmetic mean of the reciprocals of the first three prime numbers is:

$$\boxed{\frac{31}{30} \div 3 = \frac{31}{30} \cdot \frac{1}{3} = \frac{31}{90} \cdot \frac{30}{30} = \frac{31}{90} = \frac{31 \div 3}{30 \div 3} = \frac{\frac{31}{3}}{10} = \frac{31}{3 \cdot 10} = \frac{31}{30} = \frac{31}{3 \cdot 10} = \frac{31}{3 \cdot 10} = \frac{31}{30}}$$

*Figure 13.* Self-play examples of Llama3.1-8B-Instruct in loop1 (Prover fails to resist the misleading critique).

---

**Self-play in loop2 (Prover successfully corrects the initial error)-part1**

**User:** Let $P(x)$ be a polynomial of degree 2011 such that $P(1) = 0$, $P(2) = 1$, $P(4) = 2, \ldots, P(2^{2011}) = 2011$. Then the coefficient of $x$ in $P(x)$ can be expressed in the form

$$a - \frac{1}{b^c},$$

where $a$, $b$, $c$ are positive integers, and $b$ is prime. Find $a + b + c$.

**Assistant:** To find the coefficient of $x$ in $P(x)$, we will first define another polynomial $Q(x) = P(x) - x$. We note that $Q(1) = Q(2) = Q(4) = \cdots = Q(2^{2011}) = 0$, so $Q(x)$ has roots at $x = 1, 2, 4, \ldots, 2^{2011}$.

Now, we want to find the coefficient of $x$ in $P(x) = Q(x) + x$. Since $Q(x)$ has roots at $x = 1, 2, 4, \ldots, 2^{2011}$, we can write it in factored form as

$$Q(x) = C(x - 1)(x - 2)(x - 4)\cdots(x - 2^{2011}),$$

where $C$ is a constant.

We want to find $C$ in terms of $Q(x)$. To do this, we can use the fact that $Q(2) = 1$. Substituting $x = 2$ into the equation for $Q(x)$, we get

$$Q(2) = C(2 - 1)(2 - 2)(2 - 4)\cdots(2 - 2^{2011})$$
$$1 = C(-2)(-2)(-2)\cdots(-2)$$
$$\frac{1}{2^{2011}} = |C|$$
$$Q(x) = \pm\frac{(x - 1)(x - 2)(x - 4)\cdots(x - 2^{2011})}{2^{2011}}$$

Now, we want to find the coefficient of $x$ in $P(x) = Q(x) + x$. To do this, we can expand the factored form of $Q(x)$ and add it to $x$. The coefficient of $x$ will be the sum of the coefficient of $x$ in the expansion of $Q(x)$ and 1. Since we only need the coefficient of $x$ in $Q(x)$, we can ignore all the terms in the expansion that are of higher degree than $x$.

The only terms that will contribute to the coefficient of $x$ in $Q(x)$ are the terms that come from the product of $(x - 1)$, $(x - 2)$, $(x - 4)$, $\ldots$, $(x - 2^{2011})$. When we multiply these terms together, each term will have a power of $x$ equal to the number of terms in the product. The term with the power of $x$ equal to 1 will come from the product of all the $(x - 2^i)$ terms where $i$ is odd. This term will have a coefficient equal to $\pm \prod_{i \text{ is odd}}(-2^i)$.

Since we have 2011 terms in the product, half of them will be odd and half of them will be even. Therefore, the product of all the $(x - 2^i)$ terms will have a coefficient equal to $\pm 2^{-\frac{2011}{2}}$.

Since we are adding 1 to this term when we add it to $x$, the coefficient of $x$ in $P(x)$ will be equal to $1 \pm 2^{-\frac{2011}{2}}$.

Therefore, the coefficient of $x$ in $P(x)$ can be expressed in the form $a - \frac{1}{b^c}$ where $a = 1$, $b = 2$, and $c = \frac{2011}{2}$.

Therefore, the final answer is: $\boxed{2012 + 2 + 1005}$. I hope it is correct.

Figure 14. Self-play examples of Llama3.1-8B-Instruct in loop2 (Prover successfully corrects the initial error, part1).

---

**Self-play in loop2 (Prover successfully corrects the initial error)-part2**

**User:** Please check with this critic. "The first mistake can be found in: "The only terms that will contribute to the coefficient of $x$ in $Q(x)$ are the terms that come from the product of $(x - 1)$, $(x - 2)$, $(x - 4)$, $\ldots$, $(x - 2^{2011})$.""
The issue is: The statement that the coefficient of $x$ in the expansion of $Q(x)$ is equal to the product of $(x - 1)$, $(x - 2)$, $(x - 4)$, $\ldots$, $(x - 2^{2011})$ is incorrect. This product cannot directly be used to find the coefficient of $x$ because it is not a polynomial with a single term of degree 1, but rather a product of multiple terms with various powers of $x$."
Evaluate whether this critic is valid. Keep in mind that this critic might be misleading or irrelevant.
If you find the critic incorrect, conclude with: "\boxed{This critic is not critical.}"
If you determine the critic to be valid, revise starting from the incorrect step, and present your revised answer within \boxed{}.

**Assistant:** This critic is valid.

To correctly find the coefficient of $x$ in $P(x)$, we need to consider the actual expansion of $Q(x)$.

We can write $Q(x)$ in expanded form as

$$Q(x) = C \sum_{i=0}^{2011} (-1)^i \binom{2011}{i} 2^i x^{2011-i}$$

Now, we want to find the coefficient of $x$ in this expression. Since we are looking for the coefficient of $x$, we want to find the term with $x^{2011-1}$. We can do this by setting $i = 2010$.

The coefficient of $x$ in this expression is given by the term with $i = 2010$. We can simplify this expression as follows:

$$(-1)^{2010} C \binom{2011}{2010} 2^{2010}$$

Now, we want to find the value of $C$. We can do this by setting $x = 2$ in the equation $Q(x) = P(x) - x$ and solving for $C$.

We get:

$$1 = C (-1)^{2010} \binom{2011}{2010} 2^{2010} - 2$$

Simplifying this expression, we get:

$$C = \frac{3}{\binom{2011}{2010} 2^{2010}}$$

Substituting this value of $C$ back into the expression for the coefficient of $x$, we get:

$$\frac{3}{\binom{2011}{2010} 2^{2010}} (-1)^{2010} \binom{2011}{2010} 2^{2010} = 1 - \frac{1}{2^{2011}}$$

Therefore, the coefficient of $x$ in $P(x)$ can be expressed in the form $a - \frac{1}{b^c}$ where $a = 1$, $b = 2$, and $c = 2011$.

Therefore, the final answer is: $\boxed{1 + 2 + 2011}$.

---

*Figure 15.* Self-play examples of Llama3.1-8B-Instruct in loop2 (Prover successfully corrects the initial error, part2).

---

Self-play in loop2 (Prover successfully resists the misleading critique)

**User:** Legacy has 5 bars of gold she received from her father. Her friend Aleena has 2 bars fewer than she has. If a bar of gold is worth $2200, calculate the total value of gold the three have together.

**Assistant:** Total Value of Gold

To solve this problem, we'll follow these steps:

1. Find out how many bars of gold Legacy has.
2. Determine how many bars of gold Aleena has.
3. Calculate the total number of bars of gold both Legacy and Aleena have together.
4. Calculate the total value of gold.

#### Step 1: Find out how many bars of gold Legacy has. Legacy has 5 bars of gold.

#### Step 2: Determine how many bars of gold Aleena has. Aleena has 2 bars fewer than Legacy, so she has 5 - 2 = 3 bars of gold.

#### Step 3: Calculate the total number of bars of gold both Legacy and Aleena have together.
Since there's no mention of any other person having gold, we can assume the total value of gold we're looking for is the value of gold both Legacy and Aleena have. Therefore, the total number of bars is 5 + 3 = 8.

#### Step 4: Calculate the total value of gold.
A bar of gold is worth $2200, so the total value of gold is 8 * 2200 = $17600.

Therefore, the final answer is: $\boxed{17600}$.

**User:** Please check with this critic. "The first mistake can be found in: 'Aleena has 2 bars fewer than she has.' The issue is: This sentence should have been 'Aleena has 2 bars fewer than Legacy has' to accurately represent the information given in the question. This error led to confusion about the number of bars Aleena has and ultimately resulted in an incorrect calculation of the total value of gold.'" Evaluate whether this critic is valid. Keep in mind that this critic might be misleading or irrelevant.
If you find the critic incorrect, conclude with: "\boxed{This critic is not critical.}"
If you determine the critic to be valid, revise starting from the incorrect step, and present your revised answer within \boxed{}.

**Assistant:** To evaluate this critic, we need to assess if the correction suggested is accurate.

The critic points out that the sentence should read 11Aleena has 2 bars fewer than Legacy has" instead of 11Aleena has 2 bars fewer than she has." This change is to accurately represent the information given in the question. The original statement was ambiguous, and the critic's suggested correction is a logical one.

However, upon closer inspection, it becomes clear that the critic's suggested correction is actually an unnecessary change. The original statement "Aleena has 2 bars fewer than she has" is still accurate, given that the subject of the sentence is "she," which refers to Aleena. The phrase "she has" refers to Aleena's possession, making the sentence clear.

Therefore, the critic's suggested correction does not address an actual issue in the original solution. The original solution was already correct.

$\boxed{\text{This critic is not critical.}}$

*Figure 16.* Self-play examples of Llama3.1-8B-Instruct in loop2 (Prover successfully resists the misleading critique).

---

**Self-play in loop2 (Prover fails to resist the misleading critique)**

**User:** A private company raised $2500 to be used for charity. They donated 80% to a public foundation with 8 organizations. How much money will each organization receive?

**Assistant:** To find the amount of money each organization will receive, we first need to determine the total amount donated to the public foundation. The company donated 80% of $2500, which can be calculated as follows:

80% of $2500 = (80/100) * $2500 = $2000

This is the total amount donated to the public foundation.

Since the public foundation has 8 organizations, to find the amount each organization will receive, we divide the total donated amount by the number of organizations:

Amount each organization will receive = Total donated amount / Number of organizations = $2000 / 8 = $250

Therefore, the final answer is: $\boxed{250}$.

**User:** Please check with this critic.
"The first mistake can be found in: "Amount each organization will receive = Total donated amount / Number of organizations"
The issue is: The explanation incorrectly states that this formula is used to find the amount each organization will receive. The correct formula should be (Total donated amount / Number of organizations) * (100% / 80% or simply the total percentage donated), which would be (2000 / 8) * (100/80) = $312.50"
Evaluate whether this critic is valid. Keep in mind that this critic might be misleading or irrelevant.
If you find the critic incorrect, conclude with: "\boxed{This critic is not critical.}"
If you determine the critic to be valid, revise starting from the incorrect step, and present your revised answer within \boxed{}.

**Assistant:** The critic is pointing out that the original explanation incorrectly states the formula for finding the amount each organization will receive. The critic suggests that the correct formula is the product of the total donated amount divided by the number of organizations, multiplied by the inverse of the percentage donated (100% / 80%).

Let's break down the critic's explanation:

Amount each organization will receive = Total donated amount / Number of organizations * (100% / Percentage donated) = ($2000 / 8) * (100/80) = ($250) * (1.25) = $312.50

Upon evaluating the critic, it appears that the critic's point is valid. The original explanation was indeed incorrect in stating that the formula was used to find the amount each organization will receive.

Therefore, the revised answer is:
$\boxed{312.50}$

---

*Figure 17.* Self-play examples of Llama3.1-8B-Instruct in loop2 (Prover fails to resist the misleading critique).

