# OpenReview forum: "Improving Rationality in the Reasoning Process of Language Models through Self-playing Game"
_ICML.cc/2025/Conference — ICML 2025 poster_

### Official Review · Reviewer_GFAY · 2025-03-04

**Overall Recommendation:** 3

**Summary:**

The paper presents the Critic-Discernment Game (CDG), a self-play approach that enhances the reasoning of large language models (LLMs). In CDG, a "prover" generates solutions, while two critics—helpful and misleading—offer feedback. The prover learns to correct mistakes from the helpful critic and resist misleading critiques through reinforcement learning. The method is tested on tasks like mathematical reasoning, error detection, self-correction, and long-chain reasoning, showing significant improvements over baseline models. CDG enables LLMs to improve their reasoning capabilities without human supervision, demonstrating the power of self-play and reinforcement learning in model training.

**Claims And Evidence:**

Yes.

**Essential References Not Discussed:**

No.

**Experimental Designs Or Analyses:**

While the paper shows improvements on GSM8K and MATH500, one limitation is the lack of explicit testing for generalization to other problem domains or tasks outside of mathematical reasoning.

**Methods And Evaluation Criteria:**

Yes.

**Other Comments Or Suggestions:**

While the experiments are well-conducted, some aspects of the methodology (e.g., dataset creation, sampling strategies) are not presented in full detail. For example, the paper mentions the use of regular expressions and the SymPy grader for evaluating solutions, but further clarification on how these tools are integrated into the RL training process would improve transparency.

**Other Strengths And Weaknesses:**

Strength:

1.	The paper introduces a novel Critic-Discernment Game (CDG) to enhance the rationality of reasoning in LLMs through a self-play framework. While self-play has been used effectively in domains like games (e.g., AlphaGo), its application to improving error correction and long-chain reasoning in LLMs is a unique contribution.

2.	The authors demonstrate significant improvements across multiple tasks, including mathematical reasoning, stepwise error detection, self-correction, and long-chain reasoning.

3.	By incorporating a self-reflection mechanism with iterative feedback, the paper proposes a method that could significantly improve performance on challenging problems that require several reasoning steps, a major strength in advancing LLM capabilities.

Weakness:

1.	While the paper reports significant improvements on tasks related to mathematical reasoning and self-correction, the focus on these domains means the broader applicability of the CDG method remains unclear.

2.	The paper does not provide an extensive discussion on the sensitivity of the model’s performance to hyperparameters (e.g., the thresholds used in the RL setup). These hyperparameters can significantly affect the outcome of reinforcement learning-based methods, and without clear validation or tuning, the results may not be as stable or generalizable in different contexts.

3.	The training process involving multiple self-play rounds and critics is resource-intensive. While the paper demonstrates the effectiveness of CDG, it does not provide a comprehensive analysis of the computational cost or the training time involved. It would be helpful to include a comparison of the cost-benefit tradeoff between CDG and other methods, such as instruction tuning or preference optimization.

**Questions For Authors:**

Please see the weakness.

**Relation To Broader Scientific Literature:**

The key contribution of the paper is related to the following prior research:

1.	Self-play and Reinforcement Learning for Reasoning. Self-play in LLMs has been explored in other contexts, such as the Adversarial Taboo Game for improving pre-trained models’ reasoning abilities (Cheng et al., 2024b), and the Spar framework (Cheng et al., 2024a), which improves instruction-following by training with self-play.

2.	Improving Reasoning through Feedback. Process-based Reward Models (PRMs) (Zhang et al., 2025; Uesato et al., 2022a) and Step-DPO (Lai et al., 2024) also focus on improving reasoning by using feedback at the step level.

3.	Mathematical Reasoning and Long-Chain Reasoning. Math Word Problem Solving (Cobbe et al., 2021a) and Mathematical Reasoning in LLMs (Gulcehre et al., 2023) have also seen substantial improvements from techniques like process supervision and feedback-based training.  Chain-of-Thought Prompting (Wei et al., 2022) and OpenAI’s Chain-of-Thought Models (Wu et al., 2024a): These methods have been shown to improve performance on long-form reasoning tasks by encouraging models to reason through multiple steps explicitly.

**Theoretical Claims:**

No issues.

---

> ### Author Rebuttal · Authors · 2025-03-31
>
> > **Experimental Designs Or Analyses:**
> > While the paper shows improvements on GSM8K and MATH500, one limitation is the lack of explicit testing for generalization to other problem domains or tasks outside of mathematical reasoning.
> > **Weakness:**
> > While the paper reports significant improvements on tasks related to mathematical reasoning and self-correction, the focus on these domains means the broader applicability of the CDG method remains unclear.
>
> We agree that evaluating generalization beyond GSM8K and MATH is important. To this end, we conduct additional experiments on tasks outside the original training distribution, including diverse reasoning benchmarks (LogiQA, ARC-Easy, and ARC-Challenge) and more difficult mathematical problems (Minerva-MATH). Results are shown below:
>
> | Model| Minerva| LogiQA| ARC-E| ARC-C|
> |-|-|-|-|-|
> | Llama3.1-8B-Instruct| 31.24| 31.18| 79.76| 54.95|
> | + CDG| **33.10**| **32.26**| **80.05**| **55.80**|
>
> These results show that CDG yields consistent gains across different domains, suggesting it enhances general reasoning capabilities.
> We also train CDG on Dapo-17k [1], a harder dataset. Results on the challenge **AIME-24** test set are shown below:
>
> | Model| Pass@1| Pass@4| Pass@8|
> |--|--|--|--|
> | Qwen2.5-1.5B-Instruct| 1.67| 6.67| 6.67|
> | + CDG-1| 2.67| 3.33| 3.33|
> | + CDG-2| 5.33| 6.67| 6.67|
> | + CDG-3| **5.67**| **10.0**| **16.7**|
>
> We average over 10 runs due to the small test size. CDG-3 improves performance from 1.67 to 5.67 using only 18k self-generated examples. For comparison, Qwen2.5-1.5B-MATH-Instruct reaches 10.0 after 2.5M CoT-annotated examples and GRPO. This highlights CDG’s data efficiency and effectiveness on low-performing tasks.
>
> > **Weakness**
> > The paper does not provide an extensive discussion on the sensitivity of the model’s performance to hyperparameters (e.g., the thresholds used in the RL setup).
>
> The ReST approach involves very few hyperparameters and does not require a reference or critic model. Unlike PPO or DPO, where improper values of $\beta$ can lead to training collapse (e.g., duplicate or collapsed generations), ReST does not suffer from such instability. As for selecting the threshold $\tau$, we choose it based on data balance considerations and observed performance during training.
>
> > **Weakness**
> > The training process involving multiple self-play rounds and critics is resource-intensive. While the paper demonstrates the effectiveness of CDG, it does not provide a comprehensive analysis of the computational cost or the training time involved. It would be helpful to include a comparison of the cost-benefit tradeoff between CDG and other methods, such as instruction tuning or preference optimization.
>
> We appreciate the reviewer’s concern regarding computational efficiency. We compare our method with traditional RL approaches such as Expert Iteration, from both inference (rollout) and training perspectives.
>
> During the rollout stage, the prover first generate an initial solution. Based on this solution, the critic then generates multiple critiques. The prover subsequently produces multiple second-round responses conditioned on these critiques. The overall rollout cost increases by approximately 20% compared to Expert Iteration to produce the same amount of training data.
>
> During training, the additional computational cost compared to Expert Iteration comes from training the critic. The amount of training tokens for the critic is about 25% of that for the prover.
>
> Overall, while our approach involves more LLM instances, the practical computational overhead remains moderate due to reuse of generated content. In addition, our training does not require the reference model and critic model commonly used in many RL algorithms.
>
> In the experiment for Table 3, for Expert Iteration, we report the best result under equal training budgets. For Step-DPO, we follow the original 3-round setup using 10k GPT-4o-annotated step-level preferences data.
>
> > **Other Comments Or Suggestions:**
> > While the experiments are well-conducted, some aspects of the methodology (e.g., dataset creation, sampling strategies) are not presented in full detail. For example, the paper mentions the use of regular expressions and the SymPy grader for evaluating solutions, but further clarification would improve transparency.
>
> Many details of our dataset construction, sampling strategies, and hyperparameter settings are included in the appendix due to space limitations.
> Using the SymPy grader to evaluate mathematical answers is a common practice; we follow the same evaluation setup as in qwen2.5-math-instruct [2]. We will also release the detailed evaluation code.
>
> References:
> [1] Yu Q, Zhang Z, Zhu R, et al. DAPO: An Open-Source LLM Reinforcement Learning System at Scale[J]. arXiv preprint arXiv:2503.14476, 2025.
> [2] Yang A, Zhang B, Hui B, et al. Qwen2. 5-math technical report: Toward mathematical expert model via self-improvement[J]. arXiv preprint arXiv:2409.12122, 2024.

---

### Official Review · Reviewer_Mguk · 2025-03-11

**Overall Recommendation:** 3

**Summary:**

This paper introduces the Critic-Discernment Game (CDG), a self-play approach to improve reasoning in language models without human supervision. In CDG, three roles interact: a prover that solves problems, a helpful critic that identifies errors in incorrect solutions, and a misleading critic that fabricates errors in correct solutions. Through reinforcement learning, the prover learns to maintain correct answers against misleading critiques while correcting genuine errors. Experiments on mathematical reasoning, error detection, self-correction, and long-chain reasoning tasks demonstrate consistent performance improvements, in well-aligned models like LLaMA-3.1-8B-instruct, showing that this game-based approach effectively enhances reasoning capabilities beyond traditional fine-tuning methods.

**Claims And Evidence:**

One limitation is that the mechanism by which CDG improves understanding of reasoning processes is somewhat indirect - the improvements in performance are clear, but the internal changes to model reasoning are primarily inferred from these performance gains rather than directly measured or analyzed, so it's unclear that if LLMs are truly doing the reasoning and verifications or just pretending to do so.

**Essential References Not Discussed:**

No.

**Experimental Designs Or Analyses:**

The ablation studies isolate the contribution of each critic type, and comparisons with other RL methods use the same training data and budget, but simiarly to what I discussed on Claims and Evidences, one potential issue is that most evaluations assume that improving task performance equates to improving reasoning rationality, which may not always be true, but their comprehensive task suite mitigates this concern. Also the computational overhead (efficiency) problems should be discussed when comparing with other RL methods since the use of multiple LLM instances.

**Methods And Evaluation Criteria:**

Outdated and easy dataset (GSM8k) and data sizes (  200 positive and 200 negative samples) is somewhat concerning; The author should try to exploit more challenging mathematical reasoning dataset such as AIME to avoid potential data leakage to strength the arguments. Lack of concrete discussions on how the prover evaluates whether a critique is helpful or misleading.

**Other Comments Or Suggestions:**

No.

**Other Strengths And Weaknesses:**

This work uniquely combines self-play games with RL for reasoning improvement, offering a novel training paradigm beyond traditional instruction tuning and preference optimization. See above for Weakness;

**Questions For Authors:**

1: How does the approach scale with increasingly complex reasoning tasks? Is there a limit to the reasoning length where CDG becomes less effective?
2: Could you elaborate on how the prover evaluates whether a critique is helpful or misleading? Is this purely reward-driven or are there explicit criteria?
3: What specific improvements did you observe in the reasoning process itself, beyond task performance metrics?
4: How does the approach handle ambiguous problems where multiple valid reasoning paths exist?

**Relation To Broader Scientific Literature:**

Process-based reasoning improvement: Extends work by Lightman et al. (2023) and Lai et al. (2024) on stepwise supervision, but without requiring human/superior model annotations; Builds on outcome-based methods (Anthony et al., 2017; Gulcehre et al., 2023) and step-level supervision (Zhang et al., 2025; Uesato et al., 2022), but uniquely derives rewards from game rules rather than explicit supervision

**Theoretical Claims:**

The mathematical formulation in Section 3 includes game modeling, reward functions, and reinforcement learning objectives, but these are definitions rather than theorems requiring proof.

---

> ### Author Rebuttal · Authors · 2025-03-31
>
> Thanks for your constructive feedback on our paper. Our response to your questions is as follows:
>
> > **Claims And Evidence:**
> > One limitation is that the mechanism by which CDG improves understanding of reasoning processes is somewhat indirect. It's unclear that if LLMs are truly doing the reasoning and verifications or just pretending to do so.
>
> Indeed, it is difficult to directly assess whether a model reasons in a rational manner. To address this, we evaluate performance across multiple tasks and dimensions to better capture whether the model reasons more rationally.
>
> Our experimental setup is specifically designed for this purpose. For example, the error detection task uses problems the model already solves correctly. Just as human experts who understand a solution can identify and correct their own occasional mistakes, our setup tests whether the model can do the same. Failure in such cases indicates a lack of understanding of its own reasoning rather than a general capability gap.
>
> > **Methods And Evaluation Criteria:**
> > Outdated and easy dataset (GSM8k) and data sizes. The author should try to exploit more challenging mathematical reasoning dataset such as AIME.
>
> Yes, with the rapid progress of reasoning models, GSM8K and MATH are no longer sufficient for evaluating mathematical capabilities. To address this, we train CDG on Dapo-17k [1], a more challenging dataset. The results on the **AIME-24** test set are shown below:
>
> | Model| Pass@1| Pass@4| Pass@8|
> |--|--|--|--|
> | Qwen2.5-1.5B-Instruct| 1.67| 6.67| 6.67|
> | + CDG-1| 2.67| 3.33| 3.33|
> | + CDG-2| 5.33| 6.67| 6.67|
> | + CDG-3| **5.67**| **10.0**| **16.7**|
>
> We average the results over 10 runs due to the small test size. CDG-3 improves performance from 1.67 to 5.67 using only 18k self-generated examples. In comparison, Qwen2.5-1.5B-MATH-Instruct reaches 10.0 after training on 2.5M CoT-annotated examples with GRPO. This highlights the data efficiency of CDG and its effectiveness in improving initially low-performing tasks.
>
> We further evaluate the CDG-trained models in the paper on harder math tasks (Minerva-MATH) and diverse reasoning benchmarks (LogiQA, ARC-Easy/Challenge), demonstrating the generality of our method:
>
> | Model| Minerva| LogiQA| ARC-E| ARC-C|
> |--|--|--|--|--|
> | Llama3.1-8B-Instruct| 31.24| 31.18| 79.76| 54.95|
> | + CDG| **33.10**| **32.26**| **80.05**| **55.80**|
>
>
> > **Experimental Designs Or Analyses:**
> > The computational overhead (efficiency) problems should be discussed when comparing with other RL methods since the use of multiple LLM instances.
>
> We appreciate the reviewer’s concern regarding computational efficiency. We compare our method with traditional RL approaches such as Expert Iteration, from both inference (rollout) and training perspectives.
>
> During the rollout stage, the prover first generate an initial solution. Based on this solution, the critic then generates multiple critiques. The prover subsequently produces multiple second-round responses conditioned on these critiques. The overall rollout cost increases by approximately 20% compared to Expert Iteration to produce the same amount of training data.
>
> During training, the additional computational cost compared to Expert Iteration comes from training the critic. The amount of training tokens for the critic is about 25% of that for the prover.
>
> Overall, while our approach involves more LLM instances, the practical computational overhead remains moderate due to reuse of generated content. In addition, our training does not require the reference model and critic model commonly used in many RL algorithms.
>
> > **Questions For Authors**
>
> A1: Yes, as noted in Methods and Evaluation Criteria, our method significantly improves performance on AIME24, a much more challenging dataset. The average score increased from 1.67 (pre-trained) to 5.67 (CDG-trained), demonstrating stronger capabilities on complex problems. Additionally, Section 4.4 shows clear gains on long-chain CoT tasks with average reasoning lengths exceeding 1000 tokens.
>
> A2: The prompt template for determining whether a critique is helpful or misleading is detailed in the appendix. As specified, if the model deems the critic helpful, it outputs the corrected answer in the boxed section; otherwise, it returns “This critic is not critical.” Rewards are applied accordingly using a rule-based scheme.
>
> A3: Beyond performance, we observe increased self-checking behavior. For instance, the usage of the word “check” increases by 22%, and “verify” by 69%, indicating enhanced internal verification.
>
> A4: For the same question, we sample multiple solutions from the prover. In our setup, the helpful critic only engages with incorrect solutions—where the reasoning must contain errors—while the misleading critic targets correct solutions, attempting to induce mistakes.
>
> References:
> [1] Yu Q, Zhang Z, Zhu R, et al. DAPO: An Open-Source LLM Reinforcement Learning System at Scale[J]. arXiv preprint arXiv:2503.14476, 2025.

---

### Official Review · Reviewer_1Lfc · 2025-03-15

**Overall Recommendation:** 2

**Summary:**

This paper introduces a self-play reinforcement learning approach called the Critic-Discernment Game (CDG) to improve language models' reasoning capabilities. In CDG, three roles interact: a prover provides solutions to problems, a helpful critic identifies genuine errors in incorrect solutions, and a misleading critic attempts to fabricate errors in correct solutions. The prover must learn to maintain correct answers against misleading critiques while revising genuinely incorrect solutions. Through experiments on mathematical reasoning, stepwise error detection, self-correction, and long-chain reasoning tasks, the authors demonstrate that CDG training improves the rationality of well-aligned models like LLaMA-3.1-8B-Instruct and Qwen2.5-1.5B-Instruct in their reasoning processes. The method outperforms alternative approaches like Expert Iteration and Step-DPO, highlighting the potential of self-play language games as a promising training paradigm beyond instruction tuning and preference optimization.

**Claims And Evidence:**

The claims in the submission are generally supported by clear and convincing evidence through comprehensive experiments across multiple reasoning tasks. The ablation studies effectively demonstrate that both helpful and misleading critics are necessary components of CDG, while comparisons with other RL methods convincingly show CDG's advantages over alternatives. However, some claims could benefit from stronger evidence: the paper claims to improve "rationality in reasoning" but doesn't directly measure this construct beyond task performance; the improvements, while consistent, are sometimes modest (e.g., ~1.5 percentage points on GSM8K); and the long-term stability of the improvements and potential for overfitting to the game format aren't thoroughly explored, given that GSM8K and MATH are somewhat similar tasks.

**Essential References Not Discussed:**

There are two particularly essential related works not adequately cited or discussed: "Self-critiquing models for assisting human evaluators" by Saunders et al. (2022) and "LLM Critics Help Catch LLM Bugs" by McAleese et al. (2024). Besides, the authors should carefully address the similarities and differences between their approach and Kirchner et al. (2024), particularly regarding the training dynamics and how both methods aim to improve the legibility and reliability of model outputs.

**Experimental Designs Or Analyses:**

The evaluation framework is methodologically sound for measuring different aspects of reasoning. However, I identified several validity concerns: the limited dataset diversity focused only on mathematical reasoning restricts generalizability claims; there's insufficient reporting of statistical significance (e.g., standard errors); and the paper lacks proper controls for training computation across compared methods.

**Methods And Evaluation Criteria:**

The proposed methods are conceptually sound for improving reasoning, but the evaluation criteria suffer from limited scope. While the paper uses established mathematical reasoning benchmarks (GSM8K and MATH500), this narrow focus on mathematical reasoning alone raises significant concerns about generalizability. The absence of diverse reasoning datasets spanning different domains (such as commonsense reasoning, logical deduction, scientific reasoning, or counterfactual reasoning) leaves a critical gap in understanding whether CDG's benefits extend beyond mathematical problem-solving or are domain-specific.

**Other Comments Or Suggestions:**

Please see the above reviews.

**Other Strengths And Weaknesses:**

Please see the above reviews.

**Questions For Authors:**

- Do you use three separate models or a single model with different prompts for the three roles during training?
- What do you mean by “both before and after CDG training”?

**Relation To Broader Scientific Literature:**

The paper's contributions connect to several established research areas in the broader scientific literature like RLHF (Christiano et al., 2017; Ouyang et al., 2022),  self-improvement methods (Huang et al., 2023; Zelikman et al., 2022), and chain-of-Thought reasoning (Wei et al., 2022; Yao et al., 2023).

**Theoretical Claims:**

The paper does not contain formal mathematical proofs requiring verification, as it primarily presents algorithmic approaches and empirical results.

---

> ### Author Rebuttal · Authors · 2025-03-31
>
> Thanks for your constructive feedback on our paper. Our responses to your questions are as follows:
>
> > **Claims And Evidence:**
> > The paper claims to improve "rationality in reasoning" but doesn't directly measure this construct beyond task performance.
>
> As mentioned in the introduction, recent studies suggest that LLMs often rely on pattern matching rather than truly understanding their reasoning processes. However, directly assessing whether a model reasons in a rational manner is inherently difficult. To address this, we evaluate performance across multiple tasks and dimensions to better capture whether the model is reasoning more rationally.
>
> Our experimental setup is specifically designed for this purpose. For example, the error detection task is conduct on problems the model has already demonstrated proficiency in solving. Just as human experts who understand a solution can identify and correct their own occasional mistakes, our setup tests whether the model can do the same. Failure in such cases indicates a lack of understanding of its own reasoning, rather than a general capability gap.
>
> > **Claims And Evidence:**
> > The improvements, while consistent, are sometimes modest; the long-term stability of the improvements and potential for overfitting to the game format, given that GSM8K and MATH are somewhat similar tasks.
> > **Methods And Evaluation Criteria:**
> > Evaluation criteria suffer from limited scope.
>
> We agree that evaluating generalization beyond GSM8K and MATH is important. To this end, we conduct additional experiments on tasks outside the original training distribution, including diverse reasoning benchmarks (LogiQA, ARC-Easy, and ARC-Challenge) and harder mathematical problems (Minerva-MATH). Results are shown below:
>
> | Model| Minerva| LogiQA| ARC-E| ARC-C|
> |-|-|-|-|-|
> | Llama3.1-8B-Instruct| 31.24| 31.18| 79.76| 54.95|
> | + CDG| **33.10**| **32.26**| **80.05**| **55.80**|
>
> These results show that CDG yields consistent gains across different domains, suggesting it enhances general reasoning capabilities.
>
> We further evaluate CDG on more challenging mathematical problems where the base model performs poorly. Specifically, we train CDG on the Dapo-17k dataset [1] and test on the **AIME-24**benchmark. Results are averaged over 10 runs due to the small test size:
>
> | Model| Pass@1| Pass@4| Pass@8|
> |-|-|-|-|
> | Qwen2.5-1.5B-Instruct| 1.67| 6.67| 6.67|
> | + CDG-1| 2.67| 3.33| 3.33|
> | + CDG-2| 5.33| 6.67| 6.67|
> | + CDG-3| **5.67**| **10.0**| **16.7**|
>
> CDG-3 improves Pass@1 from 1.67 to 5.67 using just 18k self-generated examples. In contrast, Qwen2.5-1.5B-MATH-Instruct reaches 10.0 after training on 2.5M CoT-annotated samples with GRPO. This demonstrates CDG’s data efficiency and strong gains.
>
> > **Experimental Designs Or Analyses:**
> > There's insufficient reporting of statistical significance (e.g., standard errors); and the paper lacks proper controls for training computation across compared methods.
>
> To assess statistical significance, we conduct t-tests and report p-values in the caption of Table 1. All four tasks show significant improvements (p < 0.05).
>
> For Expert Iteration, we report the best result under an equal training budget. For Step-DPO, we follow the original 3-round setup using 10k GPT-4o-annotated step-level preference data.
>
> > **Essential References Not Discussed:**
> > There are two particularly essential related works not adequately cited or discussed.
> > Besides, the authors should carefully address the similarities and differences between Kirchner et al. (2024).
>
> Thank you for pointing this out. We acknowledge that Saunders et al. (2022) and McAleese et al. (2024) are relevant, as they also explore LLM-based self-critiquing and error detection. We will cite and discuss them in the revision.
>
> As for Kirchner et al. (2024), while both approaches involve multi-agent interaction, the objectives differs. Their goal is to improve output legibility by training a helpful prover to outperform a sneaky one. In contrast, our focus is on enhancing the model’s understanding of its own reasoning by training a prover to distinguish helpful from misleading feedback—aiming for developing rational reasoning abilities than readability. We will clarify this distinction in the revision.
>
> > **Questions For Authors:**
> > Do you use three separate models or a single model?
> > What do you mean by “both before and after CDG training”?
>
> Q1: We use three separate models during training. Prompting alone results in noticeable style differences between the helpful and misleading critics, making them too easy for the prover to distinguish.
>
> Q2: We compare two settings: (1) finetuning the base model on long-chain reasoning data (distilled from QwQ-32B-Preview), and (2) finetuning the model after CDG training using the exact same long-chain reasoning data.
>
> References:
> [1] Yu Q, Zhang Z, Zhu R, et al. DAPO: An Open-Source LLM Reinforcement Learning System at Scale[J]. arXiv preprint arXiv:2503.14476, 2025.

---

> > ### Comment · Reviewer_1Lfc · 2025-04-03
> >
> > Thank you for the response. Regarding the similarities and differences between Kirchner et al. (2024), while I understand the objective differs, the methodologies appear quite similar. Could you elaborate more on these methodological similarities?

---

> > > ### Author Response · Authors · 2025-04-03
> > >
> > > Thank you for the insightful follow-up question. While both approaches involve a prover interacting with other agents and may appear similar on the surface, the two games are fundamentally different. Below, we clarify the similarities and differences between our **Critic-Discernment Game (CDG)** and the **Prover-Verifier Game (PVG)** proposed by Kirchner et al. (2024), beyond the distinction in objectives.
> > >
> > > **Overview of Kirchner et al. (2024)**
> > >
> > > Kirchner et al. (2024) formulate a game with three roles:
> > >
> > > 1. **Helpful Prover**: A generative model aiming to produce a correct solution that is **easy for a verifier to validate**.
> > > 2. **Sneaky Prover**: A generative model aiming to generate an **incorrect solution that appears deceptively plausible**.
> > > 3. **Verifier**: A small classifier model trained to **distinguish between correct and incorrect solutions**.
> > >
> > > While both approaches adopt a game-theoretic framework with multiple agents, the rule of the game, agent behaviors are fundamentally different from our method. In CDG, the prover engages in a second-round interaction with a critic, who may either be helpful or misleading. **The prover must then decide whether to revise or maintain its original solution based on the critic’s feedback,  not just generating a verifiable answer in a single shot**.
> > >
> > > It is worth noting that the Prover-Verifier game has been proposed in previous work[1] to improve the verifiability or checkability of generated answers. In contrast, to the best of our knowledge, we are the first to propose Critic-Discernment Game in which critics with opposing goals provide feedback on a model’s solution.
> > >
> > > **Differences**
> > >
> > > 1. The two methods have different objectives.
> > > 2. The games are fundamentally different: **the game rules, as well as the agents’ tasks and behaviors, all differ.** The configuration of the role (e.g., model size, whether it acts as a generator or classifier, and the number of dialogue turns ) also differs.
> > > 3. In the Prover-Verifier Game, a **verifier (classifier)** judges whether solutions from different provers are correct, primarily to assess solution verifiability. In contrast, the Critic-Discernment Game requires the **prover itself** to evaluate the correctness of critiques directed at its own solution, thereby assessing its understanding of its own reasoning steps.
> > > 4. Kirchner et al. (2024) use PPO for optimization; we employ Rest for optimization.
> > >
> > > **Similarities**
> > >
> > > 1. Both employ **three-role games**, where each role is played by a separate model, combining competition and cooperation.
> > > 2. Both involve **iterative training** across multiple rounds to improve the performance of each agent.
> > >
> > > While both the Prover-Verifier Game and our Critic-Discernment Game leverage game-based training to enhance model capabilities, they differ significantly in terms of game rules, objectives, agent behaviors, and reward structures. The Prover-Verifier Game does not include a critic role that evaluates the internal reasoning steps of the prover, nor does the prover engage in self-assessment of its own solution during the game. Conversely, the Critic-Discernment Game does not involve a classifier to evaluate the checkability of the solution.
> > >
> > >
> > > [1] C. Anil, G. Zhang, Y. Wu, and R. Grosse. Learning to give checkable answers with prover-verifier games. arXiv preprint arXiv:2108.12099, 2021.

---

### Official Review · Reviewer_CXvh · 2025-03-17

**Overall Recommendation:** 2

**Summary:**

The paper introduces a framework that involves training three models (prover, helpful critic, misleading critic) via reinforcement learning with the goal of improving the reasoning capability of the prover model. Through the proposed training process, the prover learns to rely only on helpful feedback, and to ignore misleading feedback, thereby gaining a better notion of correct vs. incorrect reasoning. The authors conduct experiments on two math tasks, and show some improvements over the base models.

**Claims And Evidence:**

- It is unclear to me what the goal of the technique really is. The authors claim it is about teaching the models to "understand their reasoning process". The evidence provided to support this is not super convincing. First, the performance gains are quite marginal (e.g., 0.3% boost on gsm8k), and sometimes performance drops from mroe CDG iterations.

**Essential References Not Discussed:**

N/A

**Experimental Designs Or Analyses:**

- The step-wise error detection experiment is done on problems that the model can fully solve. However, in practice, we want the model to still detect errors in its reasoning on unseen problems whether it can/cannot fully solve these problems. Overall, the experiment setup seems very artificial.

**Methods And Evaluation Criteria:**

- Evaluation is done only on two math tasks, namely gsm8k and MATH. GSM8K is a very easy task, with the initial models already achieving ~90%, which might make me think that this technique only works when the base model is already good on the tasks in question. I would be more convinced of the proposed technique if it was applied to tasks where the initial performance is quite low.

- The proposed methods require sampling and training three different models, which can be quite expensive, and the helpful and misleading critics are discarded after training. Have the authors using the same model but with different roles?

**Other Comments Or Suggestions:**

N/A

**Other Strengths And Weaknesses:**

N/A

**Questions For Authors:**

- What if you use the same model for the three roles?
- How hard was hyperparameter tuning for the ReST approach? Could you elaborate on stability of the training with ReST?

**Relation To Broader Scientific Literature:**

- I think the paper's main contribution is RL-based training of self-play with three different models. The idea is neat and interesting, although the performance gains do not justify the complexity of the approach.

**Theoretical Claims:**

N/A

---

> ### Author Rebuttal · Authors · 2025-03-31
>
> We greatly appreciate your recognition of our idea and your constructive feedback. Here are our responses to your concerns:
>
> > **Claims And Evidence:**
> > The goal of the technique is unclear and the evidence provided to support this is not super convincing.
> > **Experimental Designs Or Analyses:**
> > The step-wise error detection experiment is done on problems that the model can fully solve.
>
> We discuss these two points together, as our experimental design is closely aligned with the goal of the technique. As mentioned in the introduction, recent studies have shown that LLMs lack a true understanding of their reasoning processes and instead rely primarily on probabilistic pattern matching. Our goal is to alleviate this limitation through self-play training. However, performance on a single task (e.g., math problem solving) is insufficient to determine whether a model truly understands its reasoning process. Therefore, we evaluate the model across multiple dimensions to better assess whether it reasons in a rational manner. **As shown in Table 3, our method demonstrates clear and consistent improvements across all tasks, supporting our claim.**
>
> We now clarify the motivation behind the stepwise error detection setting. Human experts, when they truly understand a solution, can identify and fix their own mistakes when such mistakes occasionally occur. Similarly, we focus on problems the model can usually solve correctly to control for knowledge or capability gaps.  If the model still fails to detect errors in such cases, it suggests a lack of understanding of its own reasoning process rather than a lack of ability. This design is well aligned with our stated goal. In contrast, Section 4.2 (self-correction) makes no such assumption and evaluates the model on general problems, where initial answers may or may not be correct—thus covering more realistic error-correction scenarios.
>
> >**Methods And Evaluation Criteria:**
> >Evaluation is done only on two math tasks, namely gsm8k and MATH. I would be more convinced of the proposed technique if it was applied to tasks where the initial performance is quite low.
>
> Yes, with the rapid progress in reasoning models, GSM8K and MATH may no longer sufficiently evaluate mathematical capabilities. To address this, we train CDG on Dapo-17k [1], a harder dataset. Results on the **AIME-24** test set are shown below:
>
> | Model| Pass@1| Pass@4| Pass@8|
> |--|--|--|--|
> | Qwen2.5-1.5B-Instruct| 1.67| 6.67| 6.67|
> | + CDG-1| 2.67| 3.33| 3.33|
> | + CDG-2| 5.33| 6.67| 6.67|
> | + CDG-3| **5.67**| **10.0**| **16.7**|
>
> We average over 10 runs due to the small test size. CDG-3 improves performance from 1.67 to 5.67 using only 18K self-generated examples. For comparison, Qwen2.5-1.5B-MATH-Instruct achieves 10.0 after 2.5M CoT-annotated examples and GRPO. This highlights CDG’s data efficiency and effectiveness on low-performing tasks.
>
> We further evaluate the CDG-trained models on harder math (Minerva-MATH) and diverse reasoning tasks (LogiQA, ARC-Easy/Challenge), demonstrating the generality of our method:
>
> | Model| Minerva| LogiQA| ARC-E| ARC-C|
> |--|--|--|--|--|
> | Llama3.1-8B-Instruct| 31.24| 31.18| 79.76| 54.95|
> | + CDG| **33.10**| **32.26**| **80.05**| **55.80**|
>
> >The helpful and misleading critics are discarded after training. Have the authors using the same model but with different roles?
>
> Yes, this is a very meaningful point. In fact, our initial design of CDG used a single model to play different roles via prompting, but we encountered two key issues:
>
> 1. The model’s responses as helpful and misleading critics differ noticeably in style and length under different prompts, making it easy for the prover to distinguish between them.
>
> 2. Training a unified model to generate misleading critiques can cause unintended side effects, such as hallucinations during regular problem solving.
>
> To avoid these issues, we adopt three separate models with non-shared parameters. This setup yields more stable and reliable gains in the prover’s reasoning ability.
>
> > How hard was hyperparameter tuning for the ReST approach? Could you elaborate on stability of the training with ReST?
>
> The ReST approach involves very few hyperparameters and does not require a reference or critic model. Unlike PPO or DPO, where improper values of $\beta$ can lead to training collapse (e.g., duplicate or collapsed generations), ReST does not suffer from such instability. As for selecting the threshold $\tau$, we manually select it based on data balance and training performance.
>
> References:
> [1] Yu Q, Zhang Z, Zhu R, et al. DAPO: An Open-Source LLM Reinforcement Learning System at Scale[J]. arXiv preprint arXiv:2503.14476, 2025.

---

### Decision · Program_Chairs · 2025-05-01

**Decision:**

Accept (poster)

**Comment:**

This paper proposes a self-play framework in which a language model (the prover) generates solutions and is then challenged by helpful or misleading critiques to enhance its reasoning robustness. Experiments show that this Critique-Driven Game (CDG) training improves models’ ability to detect, defend, and correct reasoning errors across mathematical and long-chain reasoning tasks.

While the reviewers are debating between weak accept and weak reject, I am actually leaning somewhat higher than weak accept. Reading the main concerns of the reviewers, I don't find them very substantial. Yes, the proposed method is exemplified to be effective for similar tasks, but isn't this what most machine learning does ? I find the novelty and potential improvements of the methods to be of high interest to the ICML crowd, while the concerns are typical to many other ML papers.